# MIXTURE-OF-TRANSFORMERS LEARN FASTER: A THEORETICAL STUDY ON CLASSIFICATION PROBLEMS

## ABSTRACT

Mixture-of-Experts (MoE) models improve transformer efficiency but lack a unified theoretical explanation—especially when both feed-forward and attention layers are allowed to specialize. To this end, we study the Mixture-of-Transformers (MoT), a tractable theoretical framework in which each transformer block acts as an expert governed by a continuously trained gating network. This design allows us to isolate and study the core learning dynamics of expert specialization and attention alignment. In particular, we develop a three-stage training algorithm with continuous training of the gating network, and show that each transformer expert specializes in a distinct class of tasks and that the gating network accurately routes data samples to the correct expert. Our analysis shows how expert specialization reduces gradient conflicts and makes each subtask strongly convex. We prove that the training drives the expected prediction loss to near zero in $\mathcal{O}(\log(\epsilon^{-1}))$ iteration steps, significantly improving over the $\mathcal{O}(\epsilon^{-1})$ rate for a single transformer. We further validate our theoretical findings through extensive real-data experiments, demonstrating the practical effectiveness of MoT. Together, these results offer the first unified theoretical account of transformer-level specialization and learning dynamics, providing practical guidance for designing efficient large-scale models.

## 1 INTRODUCTION

Recently, the transformer architecture (Vaswani et al., 2017) has emerged as the foundational model across a wide range of machine learning domains, including computer vision (Bi et al., 2021; Han et al., 2022; Goldblum et al., 2024), natural language processing (Kalyan et al., 2021; Tunstall et al., 2022), and speech processing (Mehrish et al., 2023; Latif et al., 2023). Despite its success, transformers often face scalability challenges and significant computational costs when applied to diverse or large-scale tasks—particularly when these tasks involve conflicting or heterogeneous feature patterns. As a result, improving the scalability and training efficiency of transformers remains a pressing and open research problem.

A prominent strategy to address these issues is to introduce Mixture-of-Experts (MoE) layers that route tokens or samples to specialized feed-forward networks (Shazeer et al., 2017; Du et al., 2022; Xue et al., 2024; Cai et al., 2024; Mu & Lin, 2025). While these approaches significantly expand model capacity, they almost always leave self-attention shared across all experts, restricting specialization to FFNs (Shazeer et al., 2017; Cai et al., 2024). Given the central role of self-attention in capturing complex dependencies and enabling the expressive power of transformers, recent works have proposed incorporating Mixture-of-attention (MoA) mechanisms into the expert architecture, demonstrating improved empirical performance (Peng et al., 2020; Zhang et al., 2022; Csordás et al., 2024). However, these efforts are almost entirely empirical. No existing theory explains how attention- and FFN-level specialization interact, or how such architectures converge during training.

We close this gap by studying the **Mixture-of-Transformers (MoT)** model, a tractable theoretical framework in which each transformer block acts as a distinct expert with its own attention and FFN layers. A gating network dynamically assigns data to specialized experts. This setting lets us isolate and analyze the joint learning dynamics of expert specialization and attention alignment—something prior work has not addressed. We summarize our main contributions as follows.

- To the best of our knowledge, this paper presents the *first theoretical analysis with benefit characterization for full-transformer specialization.* We model MoT under a general mixture-

of-classification setup. To better understand the role of the gating network in data assignment and the specialization of self-attention and feed-forward networks (FFNs), we introduce a three-stage training algorithm with continuous gating updates. In the first stage, we freeze the self-attention layers and train only the FFNs to encourage diversity across transformers. We prove that each transformer expert specializes in a distinct class of tasks, and the gating network accurately routes data samples to the correct expert. In the second stage, we freeze the FFNs and train the attention layers. We show that self-attention further reduces the training loss by extracting relevant classification signals, highlighting a key advantage of MoT over attention-absent MoE models. Finally, in the third stage, we fine-tune the FFNs to reinforce specialization.

- We provide *theoretical guarantees* on the convergence of the MoT architecture to the optimum under our three-stage training process. Specifically, we show that the expected prediction loss converges to within $\epsilon$-accuracy in $\mathcal{O}(\log(\epsilon^{-1}))$ iterations via gradient descent. Compared to existing theoretical results for multi-head transformers trained on mixture-of-classification problems Yang et al. (2025), our analysis shows that MoT significantly shortens the convergence time from $\mathcal{O}(\epsilon^{-1})$ to $\mathcal{O}(\log(\epsilon^{-1}))$. This improvement stems from expert specialization, which reduces the impact of conflicting data gradients and simplifies the classification problem faced by each transformer. As a result, MoT benefits from the strong convexity of the per-expert loss functions, enabling faster and more stable convergence.

- Finally, our *extensive real-data experiments* validate our theoretical findings and provide practical guidance for designing effective MoT systems. Interestingly, we observe that on simple datasets, a single expert suffices to solve all tasks, leading the router to direct most data to a small subset of transformers. In contrast, for more complex datasets, each transformer specializes effectively, resulting in a well-partitioned and efficient MoT system where specialization offers a clear advantage over a single multi-head transformer.

## 2 RELATED WORKS

**Transformers.** Transformers have been extensively studied since the introduction of the self-attention mechanism (Vaswani et al., 2017), focusing on improving model efficiency, generalization, and adaptability. Empirical studies have explored transformer architectures across multiple domains, including encoder-only models (e.g., BERT (Kenton & Toutanova, 2019), ALBERT (Lan et al., 2020), and DeBERTaV3 (He et al., 2023)) and encoder-decoder architectures (e.g., T5 (Raffel et al., 2020), mT5 (Chung et al., 2024), and BART (Lewis et al., 2019)). In computer vision, ViT demonstrated that treating images as sequences of patches enables transformer-based models to match CNN performance (Dosovitskiy et al., 2021; Touvron et al., 2021; Bao et al., 2021). Meanwhile, transformers have also found applications in other machine learning areas such as speech processing (Mehrish et al., 2023; Latif et al., 2023; Tang et al., 2025).

On the theoretical side, recent efforts have aimed to formally understand the capabilities and limitations of Transformer models (Li et al., 2023a; Zhang et al., 2024; Huang et al., 2024a;b; Yang et al., 2025). A popular line of work investigates in-context learning, where transformers equipped with linear or softmax attention can solve tasks such as linear regression (Zhang et al., 2024; Huang et al., 2024a), classification (Li et al., 2024), and causal inference (Nichani et al., 2024) directly from token sequences, without parameter updates. Other studies analyze how Transformers learn tasks like binary classification (Li et al., 2023a), topic modeling (Li et al., 2023b), and position-feature correlations (Huang et al., 2024b). Of particular relevance, Yang et al. (2025) designed a two-headed transformer and analyzed its three-phase training dynamics for a two-mixture classification task. However, as the task complexity increases, their analysis no longer scales, leaving open the question of how to understand the training behavior of more general Transformer-based architectures.

**Mixture-of-experts model.** To scale transformers for increasingly complex tasks, recent work has explored the integration of MoE architectures (Shazeer et al., 2017; Riquelme et al., 2021; Du et al., 2022; Xue et al., 2024; Cai et al., 2024; Mu & Lin, 2025). For instance, Riquelme et al. (2021) introduces a sparsely-gated MoE for vision Transformers, where a gating network selectively routes tokens to different experts across the batch, enabling specialization. Similarly, Xue et al. (2024) identifies a stage-wise training dynamic in MoE models, highlighting distinct functional roles of routers and experts at different phases. Some works have shown that routing strategies can significantly influence the behavior of sparse MoE models. Perturbed cosine routing has been

demonstrated to improve the statistical stability of router learning by avoiding slow parameter interactions (Nguyen et al., 2024), while balanced assignment methods such as BASE layers focus on improving load balancing in large scale sparse models (Lewis et al., 2021). Other approaches introduce stochasticity or communication aware regularization, including Gating Dropout (Liu et al., 2022) and stochastic expert activation (Zuo et al., 2021), to enhance robustness and efficiency. These works primarily study routing mechanisms, whereas our focus is to provide a theoretical understanding of how expert specialization and attention alignment emerge in Mixture-of-Transformers under a standard routing scheme. From a theoretical standpoint, Chen et al. (2022) and Li et al. (2025) emphasize the importance of expert specialization in reducing learning loss. However, MoE models typically apply mixture routing only to the FFNs within each transformer block, while the self-attention layers remain shared among all experts (Shazeer et al., 2017; Cai et al., 2024).

**Mixture-of-attention and head-switching.** Given the central role of self-attention in driving the expressive capacity of transformers, enabling attention-specific specialization is a natural next step. Some recent efforts have attempted to propose mechanisms for routing attention heads or designing head-specific switching algorithms (e.g., Peng et al. (2020); Zhang et al. (2022); Csordás et al. (2024)). In particular, Csordás et al. (2024) introduces a head-switching strategy for MoA models that improves learning speed and convergence. However, these designs remain heuristic and lack theoretical guarantees. There is no theoretical framework showing how attention specialization improves convergence or interacts with FFN specialization in MoE settings.

## 3 SYSTEM MODEL

In this section, we present our system model, which includes the distribution in the data model and the architecture of the Mixture-of-Transformers (MoT).

**Notations.** In this paper, for a vector $v$, we let $\|v\|_2$ denote its $\ell$-2 norm. For positive constant $C_1$ and $C_2$, we define $x = \Omega(y)$ if $x > C_2|y|$, $x = \Theta(y)$ if $C_1|y| < x < C_2|y|$, and $x = \mathcal{O}(y)$ if $x < C_1|y|$. We also denote by $x = o(y)$ if $x/y \to 0$. For a matrix $A$, we use $A_i$ to denote its $i$-th column.

### 3.1 DATA MODEL

We consider an $N$-mixture classification problem trained over $K$ independent data samples $\{(\boldsymbol{X}^{(k)}, y^{(k)})\}_{k=1}^{K}$, where we assume $K = \Theta(N^2)$. Let $\mathcal{C} = \{c_n\}_{n=1}^{N} \subset \mathbb{R}^d$ and $\mathcal{V} = \{v_n\}_{n=1}^{N} \subset \mathbb{R}^d$ denote the sets of class signals and classification signals, respectively. We assume the union $\mathcal{C} \cup \mathcal{V}$ forms an orthogonal set, which is commonly adopted in the prior related work (Chen et al., 2022; Huang et al., 2024a; Yang et al., 2025). For all $n \in [N]$, we assume unit form: $\|c_n\|_2 = \|v_n\|_2 = 1$. Note that our experiments do not impose these assumptions. Let $L$ denote the number of tokens per sample. Then we define the generation of each data point $(\boldsymbol{X}, y) \in \mathbb{R}^{d \times L} \times \{\pm 1\}$ as follows.

**Definition 1** ($N$-mixture of classification). *A data sample $(\boldsymbol{X}, y)$ is generated as follows.*

- *Uniformly sample $y$ and $\epsilon \in \{\pm 1\}$.*

- *Uniformly sample $n \in [N]$ and a token position $l_0 \in [L]$; set $\boldsymbol{X}_{l_0} = c_n$.*

- *Uniformly sample $l_1 \in [L] \setminus \{l_0\}$; set $\boldsymbol{X}_{l_1} = y v_n$.*

- *Uniformly sample $n' \in [N] \setminus \{n\}$ and $l_2 \in [L] \setminus \{l_0, l_1\}$; set $\boldsymbol{X}_{l_2} = \epsilon v_{n'}$.*

- *The remaining $L - 3$ token embeddings are Gaussian noise vectors, denoted by $\{\xi_i\}_{l \in [L-3]}$, that are independently drawn as $\xi_i \sim \mathcal{N}(0, \frac{\sigma_\xi^2 \mathbf{I}_d}{d})$, where $\sigma_\xi = \mathcal{O}(1)$ is an absolute constant.*

According to Definition 1, the system must predict the label associated with the classification signal $v_n$. However, the presence of an additional classification signal $v_{n'}$ with a random label $\epsilon$ can introduce confusion and mislead the prediction. To address this, the role of the class signal $c_n$ is to indicate that the correct feature is $v_n$ (i.e., the one sharing the same index), not the distractor $v_{n'}$. Moreover, the position $l_1$ of $v_n$ within $\boldsymbol{X}$ is unknown, and the attention mechanism plays a crucial role in identifying and attending to this relevant feature, thereby improving prediction performance. This data model captures key challenges of real multi-class learning: mixed informative and distractor signals, positional uncertainty, and background noise. These elements make the setting theoretically

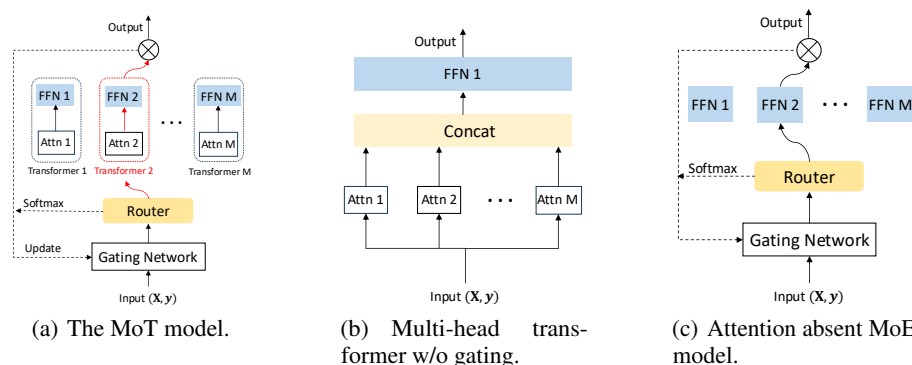

(a) The MoT model.

(b) Multi-head transformer w/o gating.

(c) Attention absent MoE model.

Figure 1: Illustrations of (a) the MoT model, (b) multi-head transformer without gating/router (Yang et al., 2025), and (c) attention absent MoE (Chen et al., 2022; Li et al., 2025).

nontrivial—allowing us to rigorously analyze expert specialization and attention alignment—while also reflecting practical scenarios such as corrupted image patches or heterogeneous tokens in NLP.

### 3.2 STRUCTURE OF THE MoT MODEL

As illustrated in Figure 1(a), the MoT model consists of a gating network, a router, and $M$ transformer experts. Each transformer comprises an attention layer followed by a feed-forward network (FFN). This structure, consistent with existing transformer studies (Yang et al., 2024; Huang et al., 2024a; Li et al., 2024), enables a tractable yet insightful understanding of the core learning dynamics in MoT, which serves as a foundational step toward understanding deeper and more complex models. Following general settings on MoE (Shazeer et al., 2017), we model the gating network to be linear with parameter matrix $\boldsymbol{\Theta} = [\boldsymbol{\theta}^{(1)} \cdots \boldsymbol{\theta}^{(M)}] \in \mathbb{R}^{d \times M}$, where $\boldsymbol{\theta}^{(i)} \in \mathbb{R}^d$ corresponds to the gating parameter for transformer $i$, for all $i \in [M]$. Given an input data pair $(\boldsymbol{X}, \boldsymbol{y})$, the gating network computes its linear output $h_i(\boldsymbol{X}; \boldsymbol{\theta}^{(i)})$ for each transformer $i$. At epoch $t$, let $\boldsymbol{h}(\boldsymbol{X}; \boldsymbol{\Theta}) := [h_1(\boldsymbol{X}; \boldsymbol{\theta}^{(1)}) \cdots h_M(\boldsymbol{X}; \boldsymbol{\theta}^{(M)})]$ denote the gating outputs for all transformer experts. As the gating is linear, we have $\boldsymbol{h}(\boldsymbol{X}; \boldsymbol{\Theta}) = \sum_{l \in [L]} \boldsymbol{\Theta}^\top \boldsymbol{X}_l$, where $\boldsymbol{X}_l$ is the $l$-th token of $\boldsymbol{X}$.

To sparsify the gating network and reduce computational cost, we adopt top-1 "switch routing", which preserves model quality while significantly lowering routing overhead, as demonstrated in prior work (Fedus et al., 2022; Chen et al., 2022; Li et al., 2025). Although this top-1 gating model is relatively simple, it is fundamental to developing a theoretical understanding of MoT. Importantly, even this simplified model presents significant theoretical challenges.

At each $t$, data point $(\boldsymbol{X}, y)$ is routed to transformer $m$ under the top-1 routing strategy as follows:
$$m = \arg\max_{i \in [M]} \{h_i(\boldsymbol{X}; \boldsymbol{\Theta}) + r^{(i)}\}, \tag{1}$$
where each $r^{(m)}$ is an independent random perturbation introduced to encourage exploration across all transformers. We show in the appendix that this routing strategy ensures stable and continuous data transitions. In addition to top-1 selection, the router also computes softmax gating probabilities,
$$\pi_i(\boldsymbol{X}; \boldsymbol{\Theta}) = \frac{\exp(h_i(\boldsymbol{X}; \boldsymbol{\theta}^{(i)}))}{\sum_{i'=1}^{M} h_{i'}(\boldsymbol{X}; \boldsymbol{\theta}^{(i')})}, \ \forall i \in [M], \tag{2}$$
which are used to update the gating network parameters $\boldsymbol{\Theta}$ in the subsequent training epoch.

After determining $m$ by Eq. (1), the router forwards the data pair $(\boldsymbol{X}, y)$ to the selected transformer expert $m$. Each transformer employs a single-layer attention mechanism whose output is defined as:
$$\boldsymbol{X}_{\mathrm{A}} = W_V^{(m)} \boldsymbol{X} \cdot \mathrm{softmax}((W_K^{(m)} \boldsymbol{X})^\top W_Q^{(m)} \boldsymbol{X}),$$
where $W_V^{(m)} \in \mathbb{R}^{d \times d}$ is the value matrix, and $W_K^{(m)}, W_Q^{(m)} \in \mathbb{R}^{d_e \times d}$ are the key and query weight matrices, respectively, for the expert $m$.

Based on the attention output $\boldsymbol{X}_A$, the FFN of transformer $m$ then computes its output as:
$$f(\boldsymbol{X}; \boldsymbol{\Theta}, \boldsymbol{w}, \boldsymbol{W}_V, \boldsymbol{W}_K, \boldsymbol{W}_Q) = \sum_{l=1}^{L} (w^{(m)})^\top W_V^{(m)} \boldsymbol{X} \cdot \mathrm{softmax}((W_K^{(m)} \boldsymbol{X})^\top W_Q^{(m)} \boldsymbol{X}_l), \tag{3}$$

where $w^{(m)} \in \mathbb{R}^d$ is the FFN weight matrix of expert $m$, and $\boldsymbol{X}_{A,l}$ is the $l$-th column of $\boldsymbol{X}_A$.

To simplify the model, we follow the prior theoretical literature on transformers (e.g, Yang et al. (2025); Huang et al. (2024a); Zhang et al. (2024)) by merging the key and query matrices $W_K^{(m)}$ and $W_Q^{(m)}$ into a single matrix $W_{KQ}^{(m)} \in \mathbb{R}^{d \times d}$ for each $m \in [M]$. Similarly, we combine the value matrix $W_V^{(m)}$ and the FFN vector $w^{(m)}$ into a single matrix $W^{(m)} \in \mathbb{R}^d$. With these simplifications, we rewrite the model output in Eq. (3) as:

$$f(\boldsymbol{X}; \boldsymbol{\Theta}, W^{(m)}, W_{KQ}^{(m)}) = \sum_{l=1}^{L} (W^{(m)})^\top \boldsymbol{X} \cdot \mathrm{softmax}(\boldsymbol{X}^\top W_{KQ}^{(m)} \boldsymbol{X}_l). \tag{4}$$

## 4 TRAINING OF THE MoT MODEL

To understand the system model introduced in Section 3, we propose a three-stage training algorithm for the MoT model. This algorithm follows the three-stage training framework previously developed for a single transformer (Yang et al., 2024; 2025), enabling us to separately examine the roles of the FFN and the attention mechanism within MoT. Under this framework, the transformers are trained in three distinct stages, while the router (i.e., the gating network) is updated continuously throughout all training epochs. In what follows, we first introduce the gradient descent (GD) update rule for the gating parameters $\boldsymbol{\Theta}_t$, and then describe the three-stage training procedure for the transformers.

**Gating parameters.** Similar to the existing MoE literature (Chen et al., 2022; Fedus et al., 2022), we first define the following empirical router loss function for training gating network parameter $\boldsymbol{\Theta}$:

$$\mathcal{L}^r(\boldsymbol{X}; \boldsymbol{\Theta}) = \frac{1}{K} \sum_{k \in [K]} \ell\Big(y^{(k)} \cdot f(\boldsymbol{X}^{(k)}; \boldsymbol{\Theta}, W^{(m_k)}, W_{KQ}^{(m_k)}) \cdot \pi_{m_k}(\boldsymbol{X}^{(k)}; \boldsymbol{\Theta})\Big), \tag{5}$$

where $\ell(z) = \log(1 + \exp(-z))$ is the logistic loss, $(\boldsymbol{X}^{(k)}, y^{(k)})$ is the $k$-th data sample, and $m_k$ is the selected expert for $\boldsymbol{X}^{(k)}$ according to Eq. (1).

Based on the router loss function defined in Eq. (5), starting from the initialization $\boldsymbol{\Theta}_0$, the gating network parameter for each expert $m$ is updated using GD:

$$\boldsymbol{\theta}_{t+1}^{(i)} = \boldsymbol{\theta}_t^{(i)} - \eta_r \cdot \nabla_{\boldsymbol{\theta}^{(i)}} \mathcal{L}^r(\boldsymbol{X}; \boldsymbol{\Theta}), \forall i \in [M], \tag{6}$$

where $\eta_r = \mathcal{O}(1)$ is the learning rate for the router. This update rule incentivizes the router to select transformer experts that yield higher values of $y^{(k)} \cdot f(\boldsymbol{X}^{(k)})$, thereby minimizing $\mathcal{L}^r(\boldsymbol{X})$ in Eq. (5).

### 4.1 THREE-STAGE TRAINING FOR TRANSFORMER EXPERTS

Let $\boldsymbol{W} = \{W^{(i)}\}_{i \in [M]}$ and $\boldsymbol{W}_{KQ} = \{W_{KQ}^{(i)}\}_{i \in [M]}$ denote the sets of neuron weights and key-query matrices for all transformer experts, respectively. At epoch $t = 0$, we initialize the neuron weights and key-query matrices of each transformer $m$ as $W_0^{(i)}, W_{KQ,0}^{(i)} \sim \mathcal{N}(0, \frac{\sigma_0^2}{d} \cdot \mathbf{I}_d)$, where $\sigma_0 = \mathcal{O}(1)$.

For each transformer, its training focuses solely on its output $f(\boldsymbol{X}^{(k)})$ with respect to any data $\boldsymbol{X}^{(k)}$ that is routed to it, aiming to align the prediction with the corresponding label $y^{(k)}$. Accordingly, at each epoch, once the router determines the routing decisions $m_k$ for all data samples, we update $W^{(i)}$ and $W_{KQ}^{(i)}$ by minimizing the following training loss for all $i \in [M]$:

$$\mathcal{L}^e(\boldsymbol{X}) = \frac{1}{K} \sum_{k=1}^{K} \ell\Big(y^{(k)} \cdot f(\boldsymbol{X}^{(k)}; \boldsymbol{\Theta}, W^{(m_k)}, W_{KQ}^{(m_k)})\Big). \tag{7}$$

In Algorithm 1, We present the three-stage training process for the MoT model based on the expert training loss defined in Eq. (7).

**Stage I: FFN specialization.** In the first stage ($t \in \{1, \cdots, T_1\}$), we fix the key-query matrices $\boldsymbol{W}_{KQ,t}$ and train only the neuron weights $\boldsymbol{W}_t$ of all Transformer experts.

Starting from the initial $\boldsymbol{W}_0$, we apply the following normalized GD update rule for each transformer $i$ at each epoch $t \in \{1, \cdots, T_1\}$ to encourage exploration and load balancing across all experts:

$$W_{t+1}^{(i)} = W_t^{(i)} - \eta \cdot \nabla_{W^{(i)}} \mathcal{L}^e(\boldsymbol{X}) / \|\nabla_{W^{(i)}} \mathcal{L}^e(\boldsymbol{X})\|_F, \tag{8}$$

where $\|\nabla_{W^{(i)}} \mathcal{L}^e(\boldsymbol{X})\|_F$ denote the Frobenius norm of $\nabla_{W^{(i)}} \mathcal{L}^e(\boldsymbol{X})$, and $\eta > 0$ is the learning rate of transformers. According to Eq. (7), for each transformer $i$, only the data samples routed to that expert contribute to the gradient in Eq. (8), i.e., any $k$ where $m_k = i$. Under this update rule, each

transformer is expected to specialize in a particular class of data. The model then proceeds to the second stage to refine the attention mechanisms for better focus on key classification signals.

**Stage II: Attention training.** In the second stage ($t \in \{T_1 + 1, \cdots, T_2\}$), we fix the neuron weights $\boldsymbol{W}$ and update only key-query matrices $\boldsymbol{W}_{KQ}$ for all transformer experts.

Starting from the initial $\boldsymbol{W}_{KQ,0}$, we apply standard GD at each epoch to update the attention layer of the selected expert $m_t$:

$$W_{KQ,t+1}^{(i)} = W_{KQ,t}^{(i)} - \eta_a \cdot \nabla_{W_{KQ}^{(i)}} \mathcal{L}^e(\boldsymbol{X}), \tag{9}$$

where $\eta_a = \mathcal{O}(1)$ is the learning rate of the attention layer, and $\mathcal{L}^e(\boldsymbol{X})$ is the training loss defined in Eq. (7). This stage allows each transformer to better focus on relevant classification features via the attention mechanism, which reduces the impact of noisy input components.

**Stage III: FFN fine-tuning.** Since the specialization learned in Stage I is preserved during attention training, the final stage ($t \in \{T_2 + 1, \cdots, T\}$) fine-tunes the neuron weights $\boldsymbol{W}$ while keeping the key-query matrices $\boldsymbol{W}_{KQ}$ fixed.

At this point, each transformer has already received sufficient updates to ensure load balancing. Hence, we switch to standard GD for an improved convergence speed:

$$W_{t+1}^{(i)} = W_t^{(i)} - \eta \cdot \nabla_{W_t^{(i)}} \mathcal{L}^e(\boldsymbol{X}). \tag{10}$$

---

**Algorithm 1** Three-Stage Training of the MoT model

1: **Input:** $T, \sigma_0, \eta_r, \eta_a, \eta, M = \Omega(N \ln(N)), T_1, T_2$;
2: Initialize $\boldsymbol{\theta}_0^{(m)} = \boldsymbol{0}, W_0^{(m)} \sim \mathcal{N}(0, \frac{\sigma_0^2}{d} \boldsymbol{I}_d)$, and
   $W_{KQ,0}^{(m)} \sim \mathcal{N}(0, \frac{\sigma_0^2}{d} \boldsymbol{I}_d), \forall m \in [M]$;
3: **for** $t = 1, \cdots, T$ **do**
4:    **for** $i = 1, \cdots, M$ **do**
5:       **if** $t \leq T_1$ **then**
6:          Update $W^{(i)}$ using Eq. (8);
7:       **else if** $t \leq T_2$ **then**
8:          Update $W_{KQ}^{(i)}$ using Eq. (9);
9:       **else**
10:         Update $W^{(i)}$ using Eq. (10);
11:       **end if**
12:    **end for**
13:    Update $\boldsymbol{\Theta}$ using Eq. (6);
14: **end for**

---

## 5 THEORETICAL RESULTS

Based on the training procedure outlined in Section 4, we first provide theoretical results that characterize the learning dynamics at each stage of training. We then analyze the convergence behavior of the MoT model under Algorithm 1, highlighting its advantages over both the attention-absent MoE model and a MoE-absent transformer baseline.

### 5.1 LEARNING DYNAMIC ANALYSIS

Before analyzing the learning dynamics, we formally define the specialization of transformer experts.

**Definition 2** (Transformer specialization). *For each transformer $i \in [M]$, let $n_i^* = \arg\max_{n \in [N]} \langle W^{(i)}, v_n \rangle$ denote the index of the signal for which the transformer is most specialized. For each class $n \in [N]$, we define its specialization expert set as $\mathcal{M}_n = \{i \in [M] | n_i^* = n\}$.*

As defined in Definition 2, the specialization of a transformer expert is determined by its FFN layer. This is because the FFN layer finalizes the classification decision, whereas the attention layer mainly contextualizes the input by capturing token relationships. Based on the definition of transformer specialization in Definition 2, we can analyze the training dynamics of the MoT model during the different stages of Algorithm 1.

The following proposition establishes the emergence of FFN specialization and router convergence by the end of Stage I.

**Proposition 1** (FFN specialization and router convergence). *Under Algorithm 1, for any epoch $t \geq T_1$, where $T_1 = \mathcal{O}(\eta^{-1}\sigma_0^{-0.5}M)$ with $\sigma_0 = \mathcal{O}(1)$ and $M = \Omega(N \log(N))$, with probability at least $1 - o(1)$, the following holds:*

$$\langle W^{(i)}, v_{n_i^*} \rangle - \langle W^{(i)}, v_n \rangle = \Theta(\sigma_0^{0.5}), \quad \forall n \neq n_i^*.$$

*Moreover, for any input $\boldsymbol{X}^{(k)}$ that includes class signal $c_{n_i^*}$, the router selects each expert $i \in \mathcal{M}_{n_i^*}$ with equal probability $\frac{1}{|\mathcal{M}_{n_i^*}|}$.*

The proof of Proposition 1 is given in Appendix D. Proposition 1 demonstrates that after $T_1$ epochs of training, each transformer's FFN layer becomes specialized to a particular class $n_i^*$, meaning that

its weight vector $W^{(i)}$ aligns significantly more with the corresponding classification signal $v_{n_i^*}$ than with any other (e.g., $c_{n_i^*}$). Intuitively, because the router assigns data randomly during this stage, each transformer receives a mixture of samples from different classes, leading to conflicting gradient directions during training. For example, a sample $(\boldsymbol{X}^{(k)}, +1)$ with $[c_n, v_n, -v_{n'}]$ pushes the gradient $\nabla_{W^{(m)}}\mathcal{L}(\boldsymbol{X})$ to align with $c_n$, while a different sample $(\boldsymbol{X}^{(k')}, -1)$ with $[c_n, -v_n, v_{n'}]$ pushes the gradient in the opposite direction, leading to $\mathbb{E}[\nabla_{W^{(m)}}\mathcal{L}^e(\boldsymbol{X}) \cdot c_n] = \mathcal{O}(\sigma_0)$. As a result, each transformer's FFN gradually shifts focus to the dominant classification signal $v_n$, enabling specialization.

Simultaneously, the router learns to forward samples within a class to the same set of specialized experts. This is because, during exploration, $\mathbb{E}[\nabla_{\boldsymbol{\theta}^{(m)}}\mathcal{L}^r(\boldsymbol{X}; \boldsymbol{\Theta}) \cdot v_n]$ also shrinks to $\mathcal{O}(\sigma_0)$ when class signals are mixed or conflicting (e.g., $(\boldsymbol{X}^{(k)}, +1)$ with $[c_n, v_n, -v_{n'}]$ and $(\boldsymbol{X}^{(k')}, -1)$ with $[c_{n'}, v_n, -v_{n'}]$ for any $n \neq n'$). As a result, the router's decisions become dominated by the clearer class signal $c_n$, ensuring that experts are specialized in that class.

In Proposition 1, $T_1$ increases with the number of transformers $M$, indicating that more transformer experts require a longer exploration period in Stage I. However, such an exploration is necessary for better learning efficiency in later stages once the specialty has been established. Note that the probability in Proposition 1 is taken over both the initialization randomness of the weight matrices, controlled by the variance $\sigma_0$, and the stochasticity inherent in expert exploration during this stage.

Building on the specialization result in Proposition 1, we now analyze the attention training phase (Stage II) to examine how the attention mechanism enhances the performance of MoT.

Let $\boldsymbol{X}(n)$ denote a feature matrix that includes the vector $c_n$. For any key vector $\mu$ and query vector $\nu$ where $\mu, \nu \in \mathcal{C} \cup \mathcal{V}$, we define the attention score between them as:

$$p_{\mu,\nu}^{(i)}(\boldsymbol{X}) := \text{softmax}(\boldsymbol{X}^\top W_{KQ}^{(i)}\mu)_{l(\nu)},$$

where $l(\nu)$ denotes the index such that $\boldsymbol{X}_{l(\nu)} = \nu$. We then characterize the learning behavior of $p_{\mu,\nu}^{(i)}(\boldsymbol{X})$ in the following proposition, demonstrating the benefit of attention in filtering out noise:

**Proposition 2** (Attention training). *Under Algorithm 1, there exists $T_2 = T_1 + \mathcal{O}(\eta_a^{-1}\sigma_0^{-0.5}N^{-1}M)$ such that for any $i \in \mathcal{M}_n$, with probability at least $1 - o(1)$, the attention score $p_{\mu,\nu}^{(i)}$ satisfies*

$$p_{\mu,\nu}^{(i)}(\boldsymbol{X}(n)) = \begin{cases} \Theta(1), & \text{if } \mu = \nu = v_n, \\ \mathcal{O}(\sigma_0), & \text{otherwise.} \end{cases}$$

The proof of Proposition 2 is given in Appendix E. This result shows that for any expert $i \in \mathcal{M}_n$ specialized in class $n$ (from Stage I), its attention layer learns to focus on the classification signal $v_n$ by assigning it a high attention weight $\Theta(1)$, while suppressing attention to irrelevant or noisy vectors with score $\mathcal{O}(\sigma_0)$. This selective focus enables MoT to further enhance its specialization by filtering out noise during inference. Similar to the analysis of Proposition 1, this attention result is also driven by training data samples containing conflicting signals. For example, data samples $(\boldsymbol{X}^{(k)}, +1)$ and $(\boldsymbol{X}^{(k')}, +1)$ with signals $[c_n, v_n, v_{n'}]$ and $[c_n, v_n, -v_{n'}]$, respectively, belong to the same class $n$ and are routed to the same set of experts in Stage II (based on Proposition 1). As a result, they drive the expected gradient direction $\mathbb{E}[\nabla_{W_{KQ}^{(i)}}\mathcal{L}^e(\boldsymbol{X}) \cdot v_{n'}]$ to decrease to $\mathcal{O}(\sigma_0)$.

In practice, we use the theoretical scaling of $T_1$ and $T_2$ as guidance for initialization and then tune them based on the observed training dynamics; we provide an empirical sensitivity study in Section B showing that MoT is robust to a broad range of choices as long as both stages are sufficiently long to allow specialization and attention alignment.

Based on Proposition 2, after $T_2$, the attention is sufficiently trained to capture the classification signals within MoT effectively. Consequently, our Algorithm 1 freezes the attention layer and proceeds to fine-tune FFN weights to better align with the current well-specialized attention patterns. We characterize the final convergence behavior of MoT under Algorithm 1 in the following theorem.

**Theorem 1** (Global Convergence of MoT). *Under Algorithm 1, for any $\epsilon > 0$, there exists $T^* = T_2 + \mathcal{O}(\log(\epsilon^{-1}))$ such that for any $t \geq T^*$, we have $\mathbb{E}[\mathcal{L}^e(\boldsymbol{X})] \leq \epsilon$. Moreover, for any $\mu \in \mathcal{V}$, the neuron weights of transformer $i \in \mathcal{M}_n$ satisfy:*

$$\langle W^{(i)}, \mu \rangle = \begin{cases} \Omega(1), & \text{if } \mu = v_n, \\ \mathcal{O}(\sigma_0), & \text{otherwise.} \end{cases} \tag{11}$$

The proof of Theorem 1 is given in Appendix F. After a period of fine-tuning in Stage III, the neuron weights of each transformer $i$ in $\mathcal{M}_n$ further amplify their response to classification signal $v_n$, leading to stronger coordination between the FFN and attention outputs. As a result, the expected test loss decreases and ultimately converges to within $\epsilon$ of the zero loss for any $\epsilon > 0$. The convergence time $\mathcal{O}(\log(\epsilon^{-1}))$ is achieved by exploiting the strong convexity of the loss function for each expert. Crucially, the specialization of transformers in MoT decomposes the original mixture-of-classification problem into simpler subtasks, each of which can be efficiently addressed by a linear FFN with its dedicated attention mechanism. In Theorem 1, we analyze the convergence of MoT using the expected gradient to maintain clarity and facilitate a direct comparison with standard multi-head transformer results in Yang et al. (2025), which are also based on expectation. We will make a detailed comparison in the following subsection.

## 5.2 Benefit Characterization of MoT

In this subsection, we characterize the benefit of MoT in comparison to two baseline architectures: standard multi-head transformers and the attention-absent MoE.

**Benefit over standard multi-head transformer:** As shown in Figure 1(b), we consider a standard transformer architecture with multi-head attention as a baseline for comparison with our MoT model. The standard transformer includes a shared, large FFN that connects all attention heads but lacks the gating mechanism and routing strategy used in MoT to assign data samples and encourage specialization across experts. As such, this architecture serves as a natural baseline for demonstrating the benefits of MoT. This baseline standard transformer has been studied recently in Yang et al. (2025), where the training dynamics for a two-headed transformer on a two-mixture classification problem have been characterized. In particular, they show that to achieve an $\epsilon$ accurate loss, i.e., $\mathbb{E}[\mathcal{L}^e(\boldsymbol{X})] \leq \epsilon$, the number of GD iterations should be at least $\mathcal{O}(\epsilon^{-1})$. In contrast, as shown in Theorem 1, our MoT takes only $\mathcal{O}(\log(\epsilon^{-1}))$ iterations (note that $T_2$ is independent of $\epsilon$), which is much faster. We will empirically validate this advantage of MoT in our experiments (Section 6).

An intuitive explanation is as follows. In a standard multi-head transformer, both the attention and FFN layers are trained on the entire dataset, which includes samples with conflicting classification signals. Although convergence is still possible, these conflicting gradients slow down the training process, resulting in a slower convergence rate of $\mathcal{O}(\epsilon^{-1})$. In contrast, our MoT model leverages expert specialization. Each transformer expert only needs to handle a subset of the data associated with a specific class and a dominant classification signal. This division reduces gradient conflict and accelerates training, leading to a faster convergence rate of $\mathcal{O}(\log(\epsilon^{-1}))$ as shown in Theorem 1.

From a more technical standpoint, the specialization of each MoT expert reduces the problem to simpler classification tasks handled by linear FFNs. This structure ensures that the loss function for each expert is strongly convex, resulting in linear convergence. In comparison, the FFN in a standard transformer must account for the mixture of all classes using nonlinear ReLU activations, leading to a generally nonconvex loss landscape and sublinear convergence behavior.

**Benefit over attention-absent MoE:** To explore the role and benefit that the attention layer plays in the model architectures, we compare our MoT with a baseline that does not contain the attention layer, i.e., each expert is simply an FFN, as depicted in Figure 1(c). We call such a baseline model as attention-absent model. Such an attention-absent model architecture has not been theoretically studied in the literature. Below we first provide the convergence guarantee for such a model.

**Lemma 1** (Convergence of Attention-Absent MoE). *Consider the MoE model without expert-specific attention layers. Then, the expected test error satisfies* $\mathbb{E}[\mathcal{L}^e(\boldsymbol{X})] \geq \Theta(\epsilon^{1-L\sigma_\xi}), \forall t \in [T]$.

The proof of Lemma 1 is given in Appendix G. Intuitively, an attention-absent MoE model is almost equivalent to the training of our MoT model in Stage I of Algorithm 1. According to Proposition 1, although the router in MoE can converge and each expert can develop its specialization, the model is unable to mitigate the negative effects caused by Gaussian noise vectors in Definition 1. As a result, the expected test error cannot be reduced below $\Theta(\epsilon^{1-L\sigma_\xi})$ with $L$ tokens per data sample. This limitation further highlights the critical role of the attention layer in the MoT model, which effectively suppresses the impact of noise and facilitates richer feature interactions across tokens within each expert. By attending to relevant feature signals, the MoT reduces the test error to an arbitrarily small value $\epsilon$ in Theorem 1. We will conduct experiments to verify the advantage of MoT over the attention-absent MoE baseline.

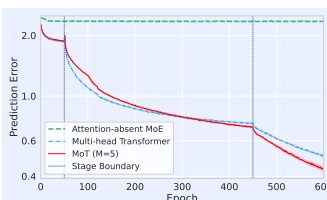 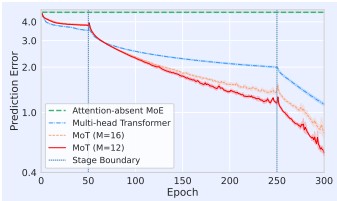

(a) Error vs. epoch on CIFAR-10.  (b) Error vs. epoch on CIFAR-100.

Figure 2: The comparison among Three-stage MoT and other baselines regarding prediction error across epochs: (1) an attention-absent MoE model with $M = 5$ experts, and (2) a multi-head transformer without gating. For CIFAR-10 (Fig. 2(a)), we set $T_1 = 50, T_2 = 450, T = 600$ and $M = 5$. For CIFAR-100 (Fig. 2(b)), we set $T_1 = 50, T_2 = 250, T = 300$, and vary $M \in \{12, 16\}$.

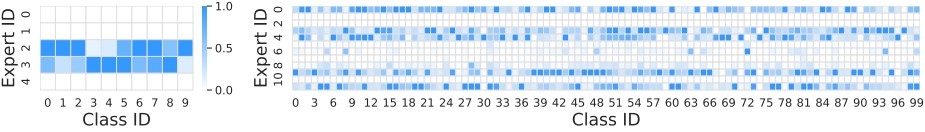

Figure 3: Routing history visualization for the MoT model on CIFAR-10 (left, epoch 580, $M = 5$) and CIFAR-100 (right, epoch 280, $M = 12$). Darker blue in each column indicates a higher proportion of samples from a class routed to that expert, showing expert specialization.

## 6 EXPERIMENT VERIFICATIONS

In this section, we conduct real-data experiments to validate our theoretical findings. We perform three-stage training following Algorithm 1 on the CIFAR-10 and CIFAR-100 image classification benchmark (Krizhevsky, 2009) using a lightweight Vision Transformer (ViT) model tailored for low-resolution inputs. The model processes $32 \times 32$ colored images by dividing them into $4 \times 4$ non-overlapping patches, yielding $L = 64$ token embeddings of dimension $d = 48$. It comprises 4 transformer encoder layers, each equipped with 4 attention heads and an FFN of width 192. We consider the setting $M = 5, T_1 = 50, T_2 = 450, T = 600$ for CIFAR-10 and $M \in \{12, 16\}, T_1 = 50, T_2 = 250, T = 300$ for CIFAR-100, respectively. Full training details and additional results are provided in Appendix B. These include comparisons between MoT and MoE with shared attention, between three-stage training and joint-training algorithms, loss sensitivity to $T_1$ and $T_2$, as well as evaluations on TinyImagenet and various NLP tasks.

We report the prediction error during training of our MoT model in Fig. 2, comparing it against two baselines: (1) standard multi-head transformer without gating/router, and (2) an MoE model with attention modules removed in the base ViT. We observe from both datasets that while the MoT model converges more slowly during Stage I, it accelerates and outperforms the multi-head transformer starting from the middle of Stage II, consistent with the theoretical insights in Theorem 1 and the benefit comparison in Section 5.2. Similarly, the MoE model without attention exhibits stagnation in training shortly after initialization, failing to improve further, which supports the error lower bound highlighted in Lemma 1. The superior performance of $M = 12$ over $M = 16$ in Fig. 2(a) indicates that fewer experts lead to faster convergence due to reduced exploration, aligning with the theoretical result that $T_1$ and $T_2$ grow with $M$ as shown in Proposition 1 and Proposition 2.

Interestingly, Fig. 2 reveals that the error gap between the MoT model and the standard multi-head transformer is substantially larger on CIFAR-100 compared to CIFAR-10. A closer look at the expert routing patterns in Fig. 3 reveals that only two of the five experts are actively utilized for CIFAR-10, whereas eight experts are engaged for CIFAR-100. This indicates a higher degree of expert specialization on the more complex CIFAR-100 dataset. These findings suggest that for simpler tasks like CIFAR-10, a standard attention-based model is often sufficient, reducing the need for expert specialization. In contrast, for more complex tasks such as CIFAR-100, the added capacity and specialization of MoT offer a clear performance advantage over standard transformers.

## 7  CONCLUSIONS AND LIMITATIONS

In this work, we introduced the first theoretical framework for Mixture-of-Transformers (MoT) to analyze full-transformer experts with both attention and FFN specialization. We developed a three-stage training algorithm and proved that MoT converges to an $\epsilon$-accurate solution in $\mathcal{O}(\log(\epsilon^{-1}))$ steps, substantially faster than the $\mathcal{O}(\epsilon^{-1})$ rate for a single transformer. Our extensive real-data experiments confirm these benefits in practice. Our analysis deliberately focuses on a simplified yet representative setting to make a full theoretical treatment possible. We assume orthogonal class and classification signals with Gaussian noise, one-layer transformer experts with merged key–query matrices, and a linear top-1 router with random exploration. These assumptions isolate the core dynamics of expert specialization and attention alignment but do not yet cover deeper architectures, nonlinear gating, or richer data correlations. Although our experiments suggest the main trends persist under more realistic conditions, extending the theory to multi-layer stacks, non-orthogonal signals, and alternative routing strategies is an important direction for future theoretical work.

## REPRODUCIBILITY STATEMENT

We have taken deliberate steps to make both our theoretical and empirical results reproducible. For our theoretical contributions, all assumptions are stated explicitly in the main text, and we have added additional discussions in Section 7. We provide complete proofs of every lemma and theorem in Appendices C–G. For our experimental results, all datasets used (CIFAR-10/100, Amazon Polarity, Yahoo Answers, Youtube Comments) are publicly available. We give a full description of pre-processing and data-splitting steps in Appendix B and have included all scripts as supplementary material. We also specify model architectures, hyperparameters, and training schedules in the main text and Appendix B. Upon acceptance we will release anonymous source code and configuration files implementing Mixture-of-Transformers, enabling independent verification of our experiments. Together, these materials satisfy the ICLR guidelines for reproducibility and allow others to reproduce our theoretical analyses and empirical findings.

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

# APPENDIX

## A   THE USE OF LARGE LANGUAGE MODELS

We utilized LLMs like ChatGPT solely to refine and enhance the readability of specific sections of this manuscript—specifically, the introduction, abstract, and several explanatory paragraphs. All research ideas, theoretical derivations, experimental designs, analyses, and conclusions are original to the authors. The LLMs were not used to generate technical content, proofs, or results. We reviewed and edited all the text produced with LLMs' assistance to ensure precision and consistency with our intended meaning.

## B   EXPERIMENT DETAILS

### B.1   HARDWARE DETAILS

The real data experiments are conducted on a system with the following hardware specifications:

- Operating system: Red Hat Enterprise Linux (RHEL) version 9
- Type of CPU: 64-Core AMD EPYC 7H12 running at 2.6 GHz
- Type of GPU: NVIDIA A100 "Ampere" with 40 GB of memory

### B.2   IMPLEMENTATION DETAILS

**Dataset.** We evaluate our Transformer models on three standard image classification datasets: CIFAR-10, CIFAR-100, and Tiny-Imagenet200.

- CIFAR-10 (Krizhevsky, 2009) contains $50,000$ $32 \times 32$ color train images and $10,000$ test images across 10 distinct object classes. Each class corresponds to a unique super-class, resulting in 10 super-classes in total.
- CIFAR-100 (Krizhevsky, 2009) contains $50,000$ $32 \times 32$ color train images and $10,000$ test images across 100 distinct object classes. They can be grouped into 20 super-classes, representing a more complex and diverse distribution of visual concepts.
- Tiny-Imagenet200 (Le & Yang, 2015) contains $100,000$ $64 \times 64$ color train images across 200 distinct object classes.

To better align our experimental setup with the theoretical data model, we apply the following modifications to CIFAR-10 and CIFAR-100: for each image, we randomly replace $15\%$ of its patches with patches from a different class, and $10\%$ with Gaussian noise. As an ablation, we also conduct the same experiment on the original CIFAR-10 and CIFAR-100 datasets, which yield results similar to those in Fig. 2.

**Training Details.** For training on CIFAR-10, CIFAR-100, and Tiny-Imagenet200, we use the AdamW optimizer under a fixed learning rate. The detailed parameters for each dataset are listed in Table 1. Additionally, we apply standard data augmentation techniques to the CIFAR-10 training data to enhance generalization. Specifically, each input image is randomly flipped horizontally with a probability of $0.5$ and slightly randomly cropped and resized to the original $32 \times 32$ resolution. No data augmentation techniques are applied during the training of CIFAR-100 and Tiny-Imagenet200.

Table 1: Training parameters for different datasets.

| Parameter | CIFAR-10 | CIFAR-100 | Tiny-Imagenet200 |
|---|---|---|---|
| Image patch unit size | 4 | 4 | 16 |
| Hidden size | 48 | 48 | 192 |
| Number of attention blocks | 4 | 4 | 3 |
| Learning Rate | 0.0008 | 0.001 | 0.0005 |
| Weight Decay | 0.01 | 0.01 | 0.05 |
| Minibatch Size | 256 | 256 | 256 |
| $T_1, T_2, T$ | $50, 450, 600$ | $50, 250, 300$ | $50, 200, 350$ |

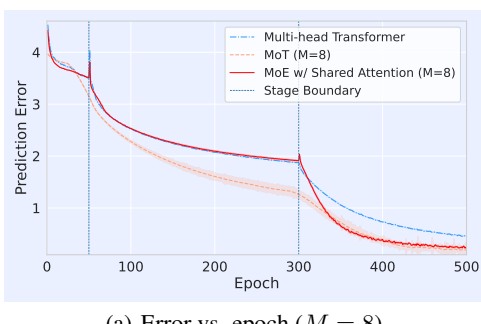 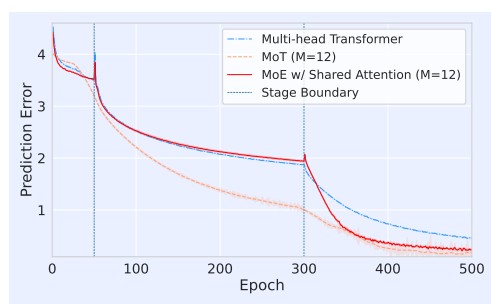

(a) Error vs. epoch ($M = 8$).          (b) Error vs. epoch ($M = 12$).

Figure 4: The comparison among Three-stage MoT, multi-head transformer, and FFN-level MoE (MoE w/ Shared Attention) regarding prediction error across epochs on CIFAR-100. We set $T_1 = 50, T_2 = 300, T = 500$. Both MoT and MoE-FFN has $M = 8$ experts (Fig. 4(a)) and $M = 12$ experts (Fig. 2(b)).

### B.3   Additional Experimental Results

**Experiments on FFN-level MoEs.** We conduct experiments on CIFAR-100 to compare MoT with standard FFN-level MoEs, which retain a shared attention backbone and only specialize the FFN layers. As shown in Fig. 4, MoT achieves better performance than FFN-level MoEs, particularly as the number of classes increases across varying expert counts ($M = 8, 12$). This supports our claim that attention specialization within experts provides a clear benefit in complex multi-task settings.

**Experiments on Joint Training MoT.** We conduct experiments on CIFAR-100 to compare 3-stage MoT with Joint Training MoT in Fig. 5. The Joint Training MoT ultimately converges to nearly the same performance point as the Multi-head Transformer, indicating that joint training could suppress expert specialization since a single transformer is powerful enough. We further track the routing history of these two methods. As is shown in Fig. 6, the Three-stage MoT achieves clear expert specialization in the third stage, showing that the staged training helps unlock meaningful specialization among experts. The reason is that, for the Joint training algorithm, in the early iterations, the gradients received by each expert are nearly indistinguishable. Without an initial mechanism to break symmetry, every expert sees almost the same mixed signal and therefore moves in the same direction. As a consequence, the router continues to dispatch inputs in a near random manner, and the system behaves similarly to a single dense transformer. This leads to a convergence rate that matches the multi-head transformer baseline rather than the improved behavior achieved when experts specialize.

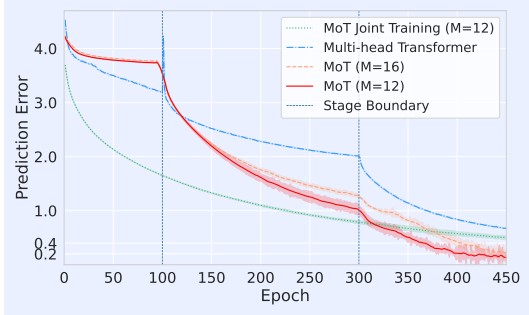

Figure 5: The comparison between Joint Training MoT ($M = 12$), Three-stage MoT ($M = 12, 16$) and multi-head transformer on CIFAR-100 dataset. We set $T_1 = 100, T_2 = 300, T = 450$.

**Experiments on Original CIFAR-10 and CIFAR-100.** We compare the training dynamics of various baselines on the original version of CIFAR-10 (Fig. 7) and CIFAR-100 (Fig. 8). As is shown in the figure, the training dynamics closely match those observed in Fig. 5, indicating that our data model assumptions hold in practice.

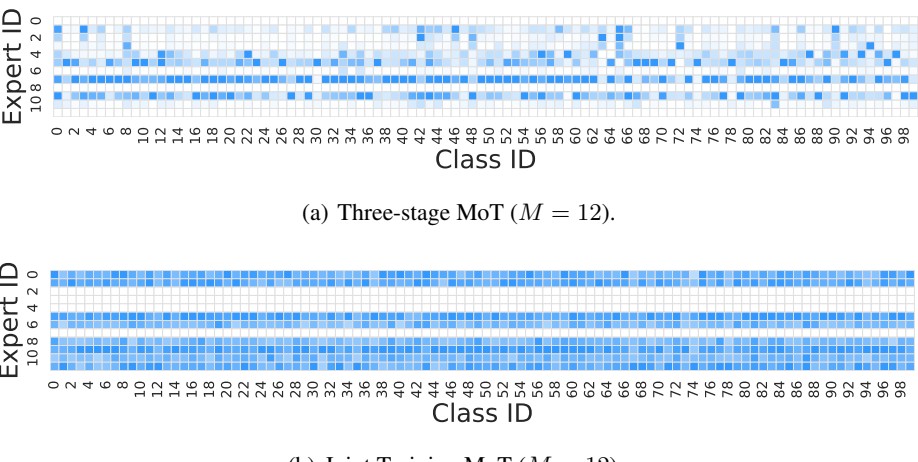

(a) Three-stage MoT ($M = 12$).

(b) Joint Training MoT ($M = 12$).

Figure 6: Routing history visualization for Three-stage MoT (Fig. 6(a)) and Joint Training MoT (Fig. 6(b)) on CIFAR-100 at epoch $400$.

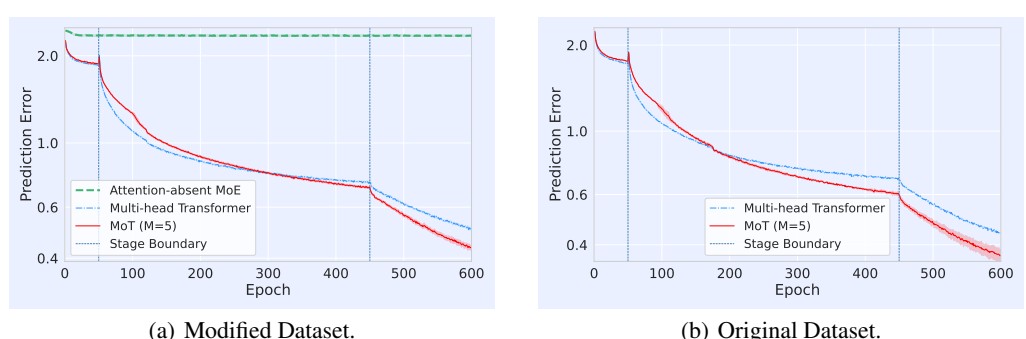

(a) Modified Dataset.                     (b) Original Dataset.

Figure 7: The comparison between Three-stage MoT ($M = 5$) and multi-head transformer on original CIFAR-10 dataset. We set $T_1 = 50, T_2 = 450, T = 600$.

**Experiments on TinyImagenet.** To deomnstrate the scalability beyond small benchmarks, we additionally conduct experiments on Tiny-Imagenet200 (Le & Yang, 2015) dataset. The training details are included in Section B.2. As is shown in Fig. 9, the MoT model outperforms the multi-head transformer starting from Stage II, mirroring the trends observed in smaller-scale experiments. These results demonstrate that our theoretical insights consistently hold as we move toward larger datasets.

**Experiments on Non-image Tasks.** In addition to the image datasets, we conduct NLP experiments to further validate our theoretical findings. Specifically, we evaluated our MoT architecture on two text classification datasets: the Amazon Polarity Classification dataset (McAuley & Leskovec, 2013; Enevoldsen et al., 2025; Muennighoff et al., 2022) (binary classification) and the Yahoo Answers dataset (Zhang et al., 2015) (10-class classification). As is shown in Figs. 10 and 11, MoT outperforms standard multi-head transformers on both datasets, particularly after sufficient attention training in Stage II (after $t = 100$). This highlights the benefit of our architecture in NLP settings. The observed trends are consistent with our vision experiments and provide further empirical support for our theoretical analysis.

We also conduct experiments on text regression using the YouTube Comments dataset (Aliak, 2025). In this task, the model is expected to predict a continuous sentiment score from YouTube comment text, reflecting the intensity and polarity of sentiment on a real-valued scale. We evaluated MoT with varying numbers of experts ($M = 3, 5$) and tracked prediction error dynamics over training. As is shown in Fig. 12, MoT begins outperforming a standard multi-head transformer from Stage II, showing consistent trends with our classification experiments and supporting the potential applicability of our theoretical findings to regression.

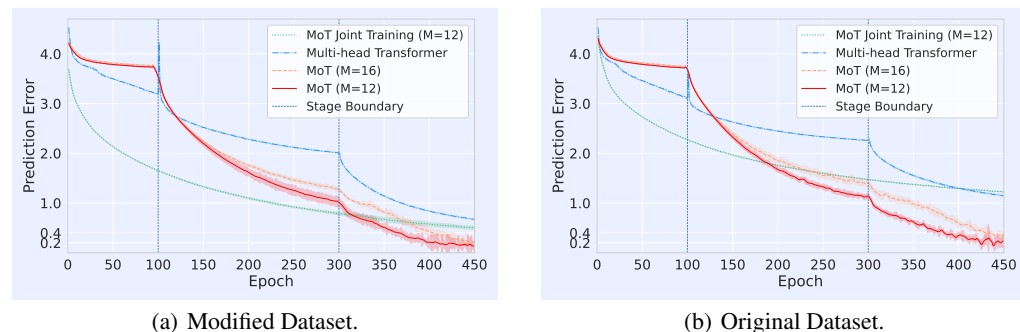

(a) Modified Dataset.                (b) Original Dataset.

Figure 8: The comparison between Joint Training MoT ($M = 12$), Three-stage MoT ($M = 12, 16$) and multi-head transformer on original CIFAR-100 dataset. We set $T_1 = 100, T_2 = 300, T = 450$.

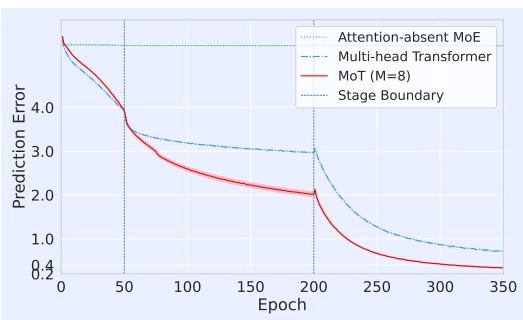

Figure 9: The comparison between Three-stage MoT ($M = 8$), multi-head transformer, and attention-absent MoE on original Tiny-Imagenet200 dataset. We set $T_1 = 50, T_2 = 200, T = 350$.

**Experiments on Sensitivity of Task Boundary Parameters.** Additionally, we evaluate the sensitivity of task boundary parameters $T_1$ and $T_2$. As shown in Fig. 13, once $T_1$ and $T_2$ are sufficiently long to train both the FFNs and attention heads, extending the training further yields only marginal performance gains, as also illustrated in Fig. 2 across all baselines.

## C    PRELIMINARY LEMMAS

We first derive the gradients of $\mathcal{L}^r$ with respect to (w.r.t) $\Theta$ and of $\mathcal{L}^e$ with respect to $W$ and $W_{KQ}$, respectively.

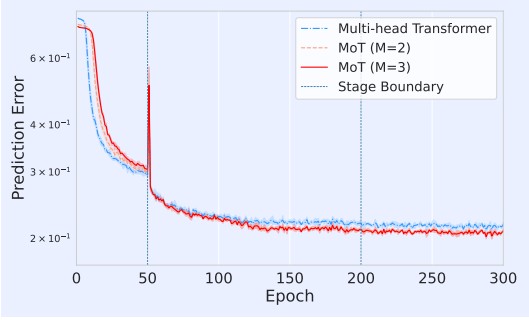

Figure 10: The comparison between Three-stage MoT ($M = 2, 3$) and multi-head transformer on Amazon Polarity Classification dataset. We set $T_1 = 50, T_2 = 200, T = 300$.

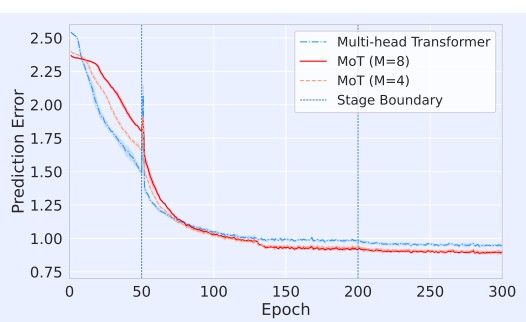

Figure 11: The comparison between Three-stage MoT ($M = 4, 8$) and multi-head transformer on Yahoo Answers dataset. We set $T_1 = 50, T_2 = 200, T = 300$.

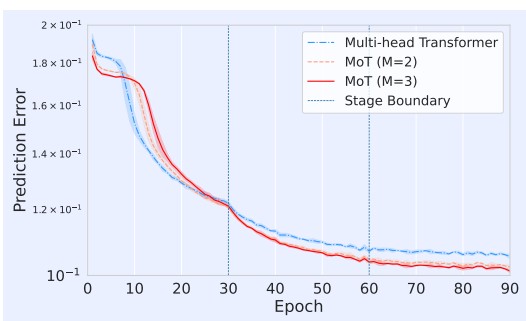

Figure 12: The comparison between Three-stage MoT ($M = 2, 3$) and multi-head transformer on Youtube Comments dataset. We set $T_1 = 30, T_2 = 60, T = 90$.

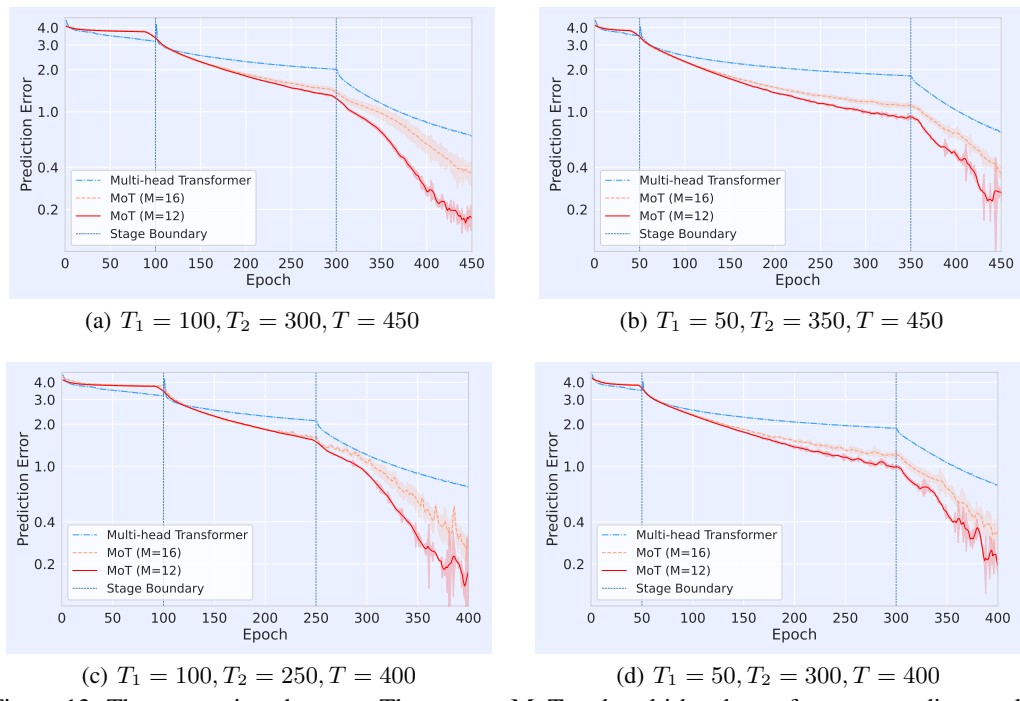

(a) $T_1 = 100, T_2 = 300, T = 450$

(b) $T_1 = 50, T_2 = 350, T = 450$

(c) $T_1 = 100, T_2 = 250, T = 400$

(d) $T_1 = 50, T_2 = 300, T = 400$

Figure 13: The comparison between Three-stage MoT and multi-head transformer regarding prediction error across epochs on CIFAR-100.

**Lemma 2** (Expression of $\nabla_{\boldsymbol{\theta}^{(i)}} \mathcal{L}^r(\boldsymbol{X}; \boldsymbol{\Theta})$). *Based on the definition of $\mathcal{L}^r(\boldsymbol{X}; \boldsymbol{\Theta})$ in eq. (5), we obtain the gradient of $\mathcal{L}^r(\boldsymbol{X}; \boldsymbol{\Theta})$ w.r.t $\boldsymbol{\Theta}$ as:*

$$\nabla_{\boldsymbol{\theta}^{(i)}} \mathcal{L}^r = \frac{1}{K} \sum_{k \in [K]} y^{(k)} \cdot f \cdot \ell' \cdot \pi_{m_k} \cdot \left( \mathbb{1}\{i = m_k\} - \pi_i \right) \cdot \sum_{l=1}^{L} \boldsymbol{X}_l^{(k)}, \tag{12}$$

*where $\mathbb{1}\{i = m_k\}$ equals 1 if $i = m_k$ and 0 otherwise, and $\ell'$ is the gradient of the logistic loss $\ell(\cdot)$.*

*Proof.* Using the chain rule, we calculate

$$\nabla_{\boldsymbol{\theta}^{(i)}} \mathcal{L}^r = \frac{1}{K} \sum_{k \in [K]} \ell' \cdot y^{(k)} \cdot \frac{\partial \pi_{m_k}}{\boldsymbol{\theta}^{(i)}}.$$

Then we calculate

$$\frac{\partial \pi_{m_k}(X; \boldsymbol{\Theta})}{\partial \boldsymbol{\theta}^{(i)}} = \begin{cases} \pi_{m_k}(X; \boldsymbol{\Theta}) \left( 1 - \pi_{m_k}(X; \boldsymbol{\Theta}) \right) \frac{\partial h_{m_k}(X; \boldsymbol{\theta}^{(i)})}{\partial \boldsymbol{\theta}^{(i)}}, & \text{if } i = m_k \\ \\ -\pi_{m_k}(X; \boldsymbol{\Theta}) \pi_i(X; \boldsymbol{\Theta}) \frac{\partial h_i(X; \boldsymbol{\theta}^{(i)})}{\partial \boldsymbol{\theta}^{(i)}}, & \text{if } i \neq m_k. \end{cases}$$

$$= \pi_{m_k} \cdot \left( \mathbb{1}\{i = m_k\} - \pi_i \right) \cdot \sum_{l=1}^{L} \boldsymbol{X}_l^{(k)}.$$

Consequently, by substituting the above expression into $\nabla_{\boldsymbol{\theta}^{(i)}} \mathcal{L}^r$, we obtain eq. Eq. (12). $\square$

**Corollary 1.** *Initially, given $\boldsymbol{\theta}^{(i)} = \mathbf{0}$ for any transformer $i \in [M]$, then we have $\sum_{i \in [M]} \boldsymbol{\theta}^{(i)} = \mathbf{0}$ at any epoch $t \in [T]$.*

*Proof.* At any epoch $t \in [T]$, we use eq. (12) to calculate

$$\sum_{i \in [M]} \nabla_{\boldsymbol{\theta}^{(i)}} \mathcal{L}^r = \frac{1}{K} \sum_{i \in [M]} \sum_{k \in [K]} y^{(k)} \cdot f \cdot \ell' \cdot \pi_{m_k} \left( \mathbb{1}\{i = m_k\} - \pi_i \right) \cdot \sum_{l=1}^{L} \boldsymbol{X}_l^{(k)}$$

$$= \frac{1}{K} \sum_{k \in [K]} y^{(k)} \cdot f \cdot \ell' \cdot \pi_{m_k} \sum_{i \in [M]} \left( \mathbb{1}\{i = m_k\} - \pi_i \right) \cdot \sum_{l=1}^{L} \boldsymbol{X}_l^{(k)}$$

$$= \frac{1}{K} \sum_{k \in [K]} y^{(k)} \cdot f \cdot \ell' \cdot \pi_{m_k} \left( 1 - \sum_{i \in [M]} \pi_i \right) \cdot \sum_{l=1}^{L} \boldsymbol{X}_l^{(k)}$$

$$= \mathbf{0},$$

where the last equality follows because $\sum_{i \in [M]} \pi_i = 1$. $\square$

**Lemma 3** (Expression of $\nabla_{\boldsymbol{W}^{(i)}} \mathcal{L}^e(\boldsymbol{X}; \boldsymbol{\Theta})$). *Based on the definition of $\mathcal{L}^e(\boldsymbol{X}; \boldsymbol{\Theta})$ in eq. (7), we obtain the gradient of $\mathcal{L}^e(\boldsymbol{X}; \boldsymbol{\Theta})$ w.r.t $\boldsymbol{W}$ as:*

$$\nabla_{\boldsymbol{W}^{(i)}} \mathcal{L}^e = \frac{1}{K} \sum_{k \in [K], m_k = i} \ell' \cdot y^{(k)} \cdot \boldsymbol{X}^{(k)} \cdot \sum_{l=1}^{L} \text{softmax}((\boldsymbol{X}^{(k)})^\top W_{KQ}^{(i)} \boldsymbol{X}_l^{(k)}). \tag{13}$$

*Proof.* At any epoch $t \in [T]$, we calculate

$$\nabla_{W^{(i)}} \mathcal{L}^e(\boldsymbol{X}) = \frac{1}{K} \sum_{k \in [K]} \ell' \cdot y^{(k)} \cdot \frac{\partial f}{\partial W^{(i)}}.$$

According to the definition of $f$ in eq. (4), for any $i \neq m_k$, we have $\frac{\partial f}{\partial W^{(i)}} = 0$. While for $i = m_k$, we have

$$\frac{\partial f}{\partial W^{(i)}} = \boldsymbol{X}^{(k)} \cdot \sum_{l=1}^{L} \text{softmax}((\boldsymbol{X}^{(k)})^\top W_{KQ}^{(i)} \boldsymbol{X}_l^{(k)}).$$

Consequently, we finally obtain $\nabla_{W^{(i)}} \mathcal{L}^e(\boldsymbol{X})$ in Eq. (13). $\square$

**Lemma 4** (Expression of $\nabla_{\boldsymbol{W}_{KQ}^{(i)}} \mathcal{L}^e(\boldsymbol{X}; \boldsymbol{\Theta})$). *Based on the definition of $\mathcal{L}^r(\boldsymbol{X}; \boldsymbol{\Theta})$ in eq. (7), we obtain the gradient of $\mathcal{L}^e(\boldsymbol{X}; \boldsymbol{\Theta})$ w.r.t $\boldsymbol{W}_{KQ}$ as:*

$$\nabla_{\boldsymbol{W}_{KQ}^{(i)}} \mathcal{L}^e = \frac{1}{K} \sum_{k \in [K], m_k = i} y^{(k)} \cdot \ell' \cdot \sum_{l=1}^{L} \boldsymbol{X}^{(k)} [\mathrm{diag}\,(\boldsymbol{p}_l^{(m_k)}) - \boldsymbol{p}_l^{(m_k)}(\boldsymbol{p}_l^{(m_k)})^\top](\boldsymbol{X}^{(k)})^\top W^{(m_k)}(\boldsymbol{X}_l^{(k)})^\top,$$

(14)

*where $\boldsymbol{p}_l^{(i)} := \mathrm{softmax}((\boldsymbol{X}^{(k)})^\top W_{KQ}^{(i)} \boldsymbol{X}_l^{(k)})$ is the $L$-dimensional vector containing the attention scores between any vector within $\boldsymbol{X}^{(k)}$ and $\boldsymbol{X}_l^{(k)}$.*

*Proof.* We define $S_l^{(i)} = \boldsymbol{X}^\top W_{KQ}^{(i)} \boldsymbol{X}_l$ to be the attention weight for expert $i$, where $S_l^{(i)} \in \mathbb{R}^L$. Then we calculate the gradient of $\boldsymbol{W}_{KQ}$ as:

$$\nabla_{\boldsymbol{W}_{KQ}} \mathcal{L}^e = \frac{1}{K} \sum_{k \in [k]} \frac{\partial \mathcal{L}_t^e}{\partial f} \cdot \sum_{l=1}^{L} \sum_{h=1}^{L} \frac{\partial f}{\partial S_{l,h}} \frac{\partial S_{l,h}^{(m_k)}}{\partial W_{KQ}^{(m_k)}}$$

$$= \frac{1}{K} \sum_{k \in [k]} y^{(k)} \cdot \ell'(y \cdot f) \cdot \sum_{l=1}^{L} \sum_{h=1}^{L} \frac{\partial f}{\partial S_{l,h}^{(m_k)}} \frac{\partial S_{l,h}^{(m_k)}}{\partial W_{KQ}^{(m_k)}},$$

where $S_{l,h}^{(m_k)}$ is the $h$-th element of $S_l^{(m_k)}$ with $h \in [L]$. Since $\boldsymbol{p}_l^{(i)} := \mathrm{softmax}(S_l^{(i)}) \in \mathbb{R}^L$, for any $i = m_k$, we further calculate

$$\frac{\partial f}{\partial S_{l,h}^{(m_k)}} = (W^{(m_k)})^\top \boldsymbol{X} \cdot \frac{\partial \boldsymbol{p}_{l,h}^{(m_k)}}{\partial S_{l,h}^{(m_k)}}.$$

Using the softmax derivative, we obtain:

$$\frac{\partial \boldsymbol{p}_{l,h}^{(i)}}{\partial S_{l,h}^{(i)}} = \begin{cases} \boldsymbol{p}_{l,h}^{(i)}(1 - \boldsymbol{p}_{l,h}^{(i)}), & \text{if } l = h, \\ -\boldsymbol{p}_{l,h}^{(i)} \boldsymbol{p}_{l,h}^{(i)}, & \text{if } l \neq h. \end{cases}$$

Additionally, we have $\frac{\partial S_{l,h}^{(i)}}{\partial W_{KQ}^{(i)}} = \boldsymbol{X}_i \boldsymbol{X}_j^\top \in \mathbb{R}^{d \times d}$.

Finally, we combine the above gradient expressions to derive:

$$\nabla_{\boldsymbol{W}_{KQ}} \mathcal{L}^e$$

$$= \frac{1}{K} \sum_{k \in [K], i = m_k} y^{(k)} \cdot \ell'(y^{(k)} \cdot f(\boldsymbol{X}^{(k)})) \cdot \sum_{l=1}^{L} \sum_{h=1}^{L} W^{(m_k)} \boldsymbol{X}_{:,h}^{(k)} \cdot \frac{\partial \boldsymbol{p}_{l,h}^{(m_k)}}{S_l^{(m_k)}} \cdot (\boldsymbol{X}_{:,h}^{(k)})^\top (\boldsymbol{X}_l^{(k)})^\top$$

$$= \frac{1}{K} \sum_{k \in [K], i = m_k} y^{(k)} \cdot \ell'(y^{(k)} \cdot f(\boldsymbol{X}^{(k)}))$$

$$\cdot \sum_{l=1}^{L} \sum_{h=1}^{L} [\boldsymbol{p}_{l,h}^{(m_k)}(\boldsymbol{X}_{:,h}^{(k)} - \sum_{r=1}^{L} \boldsymbol{p}_{l,r}^{(m_k)} \boldsymbol{X}_{:,r}^{(k)})(\boldsymbol{X}_{:,h}^{(k)})^\top W^{(m_k)}](\boldsymbol{X}_l^{(k)})^\top)$$

$$= \frac{1}{K} \sum_{k \in [K], i = m_k} y^{(k)} \cdot \ell'(y^{(k)} \cdot f(\boldsymbol{X}^{(k)}))$$

$$\cdot \sum_{l=1}^{L} \boldsymbol{X}^{(k)} [\mathrm{diag}\,(\boldsymbol{p}_l^{(m_k)}) - \boldsymbol{p}_l^{(m_k)}(\boldsymbol{p}_l^{(m_k)})^\top](\boldsymbol{X}^{(k)})^\top W^{(m_k)}(\boldsymbol{X}_l^{(k)})^\top,$$

where $\boldsymbol{X}_{:,h} \in \mathbb{R}^{1 \times d}$ is the $h$-th row of $\boldsymbol{X}$ and $\mathrm{diag}(\boldsymbol{p}_l^{(m_k)})$ is the Jacobian identity of $\boldsymbol{p}_l^{(m_k)}$. $\square$

We next provide useful properties of the data distribution and our MoT model. Fix transformer expert $i \in [M]$, and recall that its optimal class $n_i^*$ is defined as: $n_i^* = \arg\max_k \langle v_n, W^{(i)} \rangle$.

**Lemma 5** (Initialization). *At initial epoch $t = 0$, with probability at least $1 - o(1)$ the following properties hold for all $k \in [K]$,*

$$|\langle W_0^{(i)}, \boldsymbol{X}^{(k)} \rangle| \leq \mathcal{O}(\sigma_0),$$

$$|\nu^\top W_{KQ}^{(i)}\mu| = \mathcal{O}(\sigma_0), \forall \nu, \mu \in \mathcal{C} \cup \mathcal{V}.$$

*Proof.* Let $\delta = o(1)$ be a fixed small constant. Recall the Gaussian tail bound that for $X \sim \mathcal{N}(0, \sigma^2)$,

$$\mathbb{P}(|X| \geq t) \leq 2\exp\left(-\frac{t^2}{2\sigma^2}\right).$$

At initialization, $w_0^{(i)} \sim \mathcal{N}(0, \frac{\sigma_0^2}{d}I_d)$. For any unit vector $\nu \in \mathcal{C} \cup \mathcal{V}$, $\langle w_0^{(i)}, \nu \rangle \sim \mathcal{N}(0, \sigma_0^2/d)$.

Set $\omega = \sigma_0\sqrt{2\log(2/\delta)/d}$. Then,

$$\mathbb{P}\left(\left|\langle w_0^{(i)}, v\rangle\right| \geq \omega\right) \leq \delta.$$

That is, with probability at least $1 - \delta$, we have $|\langle w_0^{(i)}, v\rangle| \leq \sigma_0\sqrt{2\log(2/\delta)/d} = \mathcal{O}(\sigma_0)$.

Each entry of $W_{KQ,0}^{(i)}$ is i.i.d. $\mathcal{N}(0, \frac{\sigma_0^2}{d})$. For any normal vectors $\nu, \mu \in \mathbb{R}^d$, the random variable $S = \nu^\top W_{KQ,0}^{(i)}\mu$ is Gaussian with mean zero and variance $\text{Var}(S) = \frac{\sigma_0^2}{d}$. By the same tail bound, for $\omega = \sigma_0\sqrt{2\log(2/\delta)/d}$, we have $\mathbb{P}(|S| \geq \omega) \leq \delta$. Thus, with probability at least $1 - \delta$,

$$|\nu^\top W_{KQ,0}^{(i)}\mu| \leq \sigma_0\sqrt{2\log(2/\delta)/d} = \mathcal{O}(\sigma_0).$$

This completes the proof of Lemma 5. $\qquad\square$

**Lemma 6.** *Given $|\nu^\top W_{KQ}^{(i)}\mu| = \mathcal{O}(\sigma_0)$ for any $\nu, \mu \in \mathcal{C} \cup \mathcal{V}$, for $\nu_2 \neq \nu_1$, we have*

$$|\text{softmax}(\boldsymbol{X}W_{KQ}^{(i)}\mu)_{l(\nu_1)} - \text{softmax}(\boldsymbol{X}W_{KQ}^{(i)}\mu)_{l(\nu_2)}| = \mathcal{O}(\sigma_0).$$

*Proof.* For any $\nu_2 \neq \nu_1$, we calculate

$$|\text{softmax}(\boldsymbol{X}W_{KQ}^{(i)}\mu)_{l(\nu_1)} - \text{softmax}(\boldsymbol{X}W_{KQ}^{(i)}\mu)_{l(\nu_2)}|$$

$$= \text{softmax}(\boldsymbol{X}W_{KQ}^{(i)}\mu)_{l(\nu_2)}|\exp\left((\boldsymbol{X}W_{KQ}^{(i)}\mu)_{l(\nu_2)} - (\boldsymbol{X}W_{KQ}^{(i)}\mu)_{l(\nu_2)}\right) - 1|.$$

By applying Taylor series, with sufficiently small $\sigma_0$, we obtain

$$|\text{softmax}(\boldsymbol{X}W_{KQ}^{(i)}\mu)_{l(\nu_1)} - \text{softmax}(\boldsymbol{X}W_{KQ}^{(i)}\mu)_{l(\nu_2)}|$$

$$= \text{softmax}(\boldsymbol{X}W_{KQ}^{(i)}\mu)_{l(\nu_2)}\Bigg(\left|(\boldsymbol{X}W_{KQ}^{(i)}\mu)_{l(\nu_2)} - (\boldsymbol{X}W_{KQ}^{(i)}\mu)_{l(\nu_2)}\right|$$

$$+ \mathcal{O}\left(|(\boldsymbol{X}W_{KQ}^{(i)}\mu)_{l(\nu_2)} - (\boldsymbol{X}W_{KQ}^{(i)}\mu)_{l(\nu_2)}|^2\right)\Bigg)$$

$$= \mathcal{O}(\sigma_0),$$

where the last equality follows because $\text{softmax}(\boldsymbol{X}W_{KQ}^{(i)}\mu)_{l(\nu_2)} = \Theta(1)$. $\qquad\square$

Note that in the above Taylor expansion, if $|\nu_1^\top W_{KQ}^{(i)}\mu - \nu_2^\top W_{KQ}^{(i)}\mu| = \Omega(\sigma_0)$, we have

$$|\text{softmax}(\boldsymbol{X}W_{KQ}^{(i)}\mu)_{l(\nu_1)} - \text{softmax}(\boldsymbol{X}W_{KQ}^{(i)}\mu)_{l(\nu_2)}| = \Omega(\sigma_0).$$

We next propose the following lemma to show the specialization of each transformer's FFN at the initial state.

**Lemma 7** (Initial Specialization). *For $M \geq \Theta(N\log(N/\delta))$, with probability at least $1 - \delta$, we have*

$$\max_{n \neq n_i^*}\langle W_0^{(i)}, v_n\rangle \leq \left(1 - \frac{\delta}{3MN^2}\right)\langle W_0^{(i)}, v_{n_i^*}\rangle.$$

*Proof.* Recall that $W^{(i)} \sim \mathcal{N}(0, \frac{\sigma_0^2}{d}\boldsymbol{I}_d)$. Given an expert $i \in [M]$, we have $\{\langle W_0^{(i)}, v_n\rangle | n \in [N]\}$ are independent and individually drawn from $\mathcal{N}(0, \sigma_0^2)$. Let $\rho$ be the PDF of $\mathcal{N}(0, \sigma_0^2)$, and let $\Psi$ be the CDF of $\mathcal{N}(0, 1)$. Take $\Delta = \frac{\delta}{3MN^2}$, and define the non-increasing sequence of $\boldsymbol{V} = \{\langle W_0^{(i)}, v_n\rangle\}_{n=1}^N$

as $V_1 \geq \cdots \geq V_N$. Then we have

$$\mathbb{P}\left(\max_{n \neq n_i^*} \langle W_0^{(i)}, v_n \rangle \leq \left(1 - \Delta\right) \langle W_0^{(i)}, v_{n_i^*} \rangle\right)$$

$$= \int_{V_1 \geq \cdots \geq V_N} \mathbb{1}(V_2 \geq (1 - \Delta)V_1)N! \prod_n \rho(V_n) d\boldsymbol{V}$$

$$= \int_{V_1 \geq V_2} \mathbb{1}(V_2 \geq (1 - \Delta)V_1)N(N-1)\rho(V_1)\rho(V_2)\Psi(V_2)^{N-2}dV_1 dV_2$$

$$\leq \int_{V_1 \geq V_2} \mathbb{1}(V_2 \geq (1 - \Delta)V_1)N(N-1)\rho(V_1)\frac{1}{\sqrt{2\pi}}dV_1 dV_2$$

$$= \int_{V_1 \geq 0} \frac{\Delta N(N-1)}{\sqrt{2\pi}}V_1 \rho(V_1)dV_1$$

$$\leq \Delta N^2 = \frac{\delta}{3M}.$$

Consequently, we have probability of at least $1 - \delta$ such that $\max_{n \neq n_i^*} \langle W_0^{(i)}, v_n \rangle \leq \left(1 - \frac{\delta}{3MN^2}\right) \langle W_0^{(i)}, v_{n_i^*} \rangle$. $\qquad \square$

As shown in Lemma 7, each expert's specialization arises from the initialization of its weight matrices. Since these initializations are independent for different experts, the resulting specializations are independent across experts as well. In the next lemma, we prove that given a large number of experts, the MoT model ensures that there is at least one expert that specializes in learning each data class.

**Lemma 8** (Expert Specialization Guarantee). *For $M \geq \Theta(N \log(N/\delta))$, with probability at least $1 - \delta$, we have $|\mathcal{M}_k| \geq 1$ for all $k \in [K]$.*

*Proof.* If $M > N$, by the symmetric property, we have that for all $n \in [N], i \in [M]$,

$$\mathbb{P}(i \in \mathcal{M}_n) = \frac{1}{N}.$$

Therefore, the probability that $|\mathcal{M}_n|$ at least includes one expert is

$$\mathbb{P}(|\mathcal{M}_n| \geq 1) = 1 - \left(1 - \frac{1}{N}\right)^M.$$

By applying union bound, we obtain

$$\mathbb{P}(|\mathcal{M}_n| \geq 1, \forall n) = \left(1 - \left(1 - \frac{1}{N}\right)^M\right)^N \geq 1 - N\left(1 - \frac{1}{N}\right)^M \geq 1 - N \exp\left(-\frac{M}{N}\right) \geq 1 - \delta,$$

where the second inequality follows because $(1 - N^{-1})^M$ is small enough, and the last inequality follows because $M = \Omega\left(N \ln\left(\frac{N}{\delta}\right)\right)$. $\qquad \square$

Let $\boldsymbol{h} \in \mathbb{R}^M$ and $\hat{\boldsymbol{h}} \in \mathbb{R}^M$ be the output of the gating network and $\{r^{(i)}\}_{m=1}^M$ be the noise independently drawn from $\mathcal{D}_r$. Let $\boldsymbol{p}$ and $\hat{\boldsymbol{p}}$ denote the set of probabilities that experts are routed under gating outputs $\boldsymbol{h}$ and $\hat{\boldsymbol{h}}$, respectively. Here $\boldsymbol{p}^{(i)} = \mathbb{P}(\arg\max_{i' \in [M]}\{h_{i'} + r^{(i')}\} = i)$ and $\hat{\boldsymbol{p}}^{(i)} = \mathbb{P}(\arg\max_{i' \in [M]}\{\hat{h}_{i'} + r^{(i')}\} = i)$. Then we adopt an lemma from Chen et al. (2022) to capture the smoothness of the router under the random noise below.

**Lemma 9** (Smoothed Router Chen et al. (2022)). *Suppose that the probability density function of $\mathcal{D}_r$ is bounded by $\kappa$, then we have $\|\boldsymbol{p} - \hat{\boldsymbol{p}}\| \leq (\kappa M)^2 \|\boldsymbol{h} - \hat{\boldsymbol{h}}\|_\infty$.*

The proof of Lemma 9 is in Appendix C of Chen et al. (2022) and we skip it here. Lemma 9 guarantees that even with a random noise $r^{(i)}$ for each transformer $i \in [M]$, the routing decisions are still determined by the gating output $h_i$.

**Lemma 10.** *Under our data distribution in Definition 1, we have $K = \Theta(N^2)$.*

*Proof.* Each data sample is determined by a choice of $(n, n')$ with $n \in [N]$ and $n' \in [N] \setminus \{n\}$, corresponding to the class and the distractor. Thus, the total number of mixture types is

$$K = 2 \cdot 2 \cdot N \cdot (N-1) = \Theta(N^2).$$

The remaining randomness—such as positions of tokens, the signs $y$ and $\epsilon$, and Gaussian noise—does not increase the number of fundamental mixture types, only the number of observed samples per type. Thus, the number of distinct mixture types is $\Theta(N^2)$. □

# D    PROOF OF PROPOSITION 1

**Lemma 11.** *At any epoch $t \in \{1, \cdots, T_1\}$, we have $f(\boldsymbol{X}; \boldsymbol{\Theta}, W^{(m)}, W_{KQ}^{(m)}) = (\sigma_0^{0.5})$.*

*Proof.* According to Algorithm 1, in Stage I, we fix the attention layer and only update $W^{(m)}$ at each epoch. First, we have that $\mathrm{softmax}(\boldsymbol{X}^\top W_{KQ}^{(m)} \boldsymbol{X}_l) \leq 1$ and $\boldsymbol{X} = \mathcal{O}(1)$ always hold. Then, based on the normalized GD update rule of $W^{(i)}$ in eq. (8), we have

$$W_{T_1}^{(i)} = W_0^{(i)} - \mathcal{O}(\eta T_1) = \mathcal{O}(\sigma_0^{0.5}),$$

due to the fact that $W_0^{(i)} = \mathcal{O}(\sigma_0)$ derived in Lemma 5 and $T_1 = \mathcal{O}(\sigma_0^{0.5})$.

Consequently, we have $(W^{(i)})^\top \boldsymbol{X} = \mathcal{O}(\sigma_0^{0.5})$. □

For $(\boldsymbol{X}, y)$ drawn from Definition 1, we assume that the first token is class signal $\boldsymbol{X}_1 = c_n$, the second token is classification signal $\boldsymbol{X}_2 = v_n$, and the third token is classification noise $\boldsymbol{X}_3 = v_{n'}$. The other $L - 3$ tokens are Gaussian noises. Therefore, we rewrite $\boldsymbol{X} = [c_n, yv_n, \epsilon v_{n'}, \boldsymbol{\xi}]$, where $\boldsymbol{\xi} = [\xi_1, \cdots, \xi_{L-3}]$ is a Gaussian matrix. Let $\boldsymbol{X} \in \Omega_n$ if $\boldsymbol{X}_1 = c_n$, and let $\boldsymbol{X} \in \Omega_{n,n'}$ if $\boldsymbol{X}_1 = c_n$ and $\boldsymbol{X}_3 = v_{n'}$ or $\boldsymbol{X}_3 = -v_{n'}$. Furthermore, we let $\boldsymbol{X} \in \Omega_{+n,+n'}$ if $\boldsymbol{X}_2 = +v_n$ and $\boldsymbol{X}_3 = +v_{n'}$.

**Lemma 12.** *At any epoch $t \in \{1, \cdots, T_1\}$, the following properties hold:*

$$\mathbb{E}[\langle \nabla_{\boldsymbol{\theta}^{(i)}} \mathcal{L}^r, c_n \rangle] = \mathcal{O}(\frac{\sigma_0^{0.5}}{M^2}),$$

$$\mathbb{E}[\langle \nabla_{\boldsymbol{\theta}^{(i)}} \mathcal{L}^r, v_n \rangle] = \mathcal{O}(\frac{\sigma_0^{0.5}}{N^2 M^2}),$$

$$\mathbb{E}[\langle \nabla_{\boldsymbol{\theta}^{(i)}} \mathcal{L}^r, \xi_l \rangle] = \mathcal{O}(\frac{\sigma_\xi}{M^2}), \forall l \in [L - 3].$$

*Proof.* Based on the expression of $\nabla_{\boldsymbol{\theta}^{(i)}} \mathcal{L}^r$ in Eq. (12), we first calculate $\mathbb{E}[\langle \nabla_{\boldsymbol{\theta}^{(i)}} \mathcal{L}^r, c_n \rangle]$ below:

$$\mathbb{E}[\langle \nabla_{\boldsymbol{\theta}^{(i)}} \mathcal{L}^r, c_n \rangle] = \mathbb{P}(m_k = i) \frac{1}{K} \sum_{k \in [K]} y^{(k)} \cdot f \cdot \ell' \cdot \pi_{m_k} \cdot (1 - \pi_{m_k}) \cdot \sum_{l=1}^{L} (\boldsymbol{X}_l^{(k)})^\top \cdot c_n$$

$$- \sum_{i \neq m_k} \mathbb{P}(m_k = j | j \neq i) \frac{1}{K} \sum_{k \in [K]} y^{(k)} \cdot f \cdot \ell' \cdot \pi_{m_k} \cdot (-\pi_i) \cdot \sum_{l=1}^{L} (\boldsymbol{X}_l^{(k)})^\top \cdot c_n,$$

which contains two cases: (1) $m_k = i$ and (2) $m_k = j \neq i$. Given $\|c_n\|_2 = 1$, we have

$$\sum_{l=1}^{L} (\boldsymbol{X}_l^{(k)})^\top \cdot c_n = \Theta(1).$$

Additionally, we have

$$\mathbb{P}(m_k = i) = \mathbb{P}(m_k = j) = \Theta(\frac{1}{M}), \quad \pi_{m_k} = \Theta(\frac{1}{M}), \quad \pi_i = \mathcal{O}(\frac{1}{M}), \quad 1 - \pi_{m_k} = \Theta(1),$$

$$y^{(k)} = \Theta(1), \quad \ell' = \mathcal{O}(1)$$

and $f = \mathcal{O}(\sigma_0^{0.5})$ (by Lemma 11). Consequently, we have

$$\mathbb{E}[\langle \nabla_{\boldsymbol{\theta}^{(i)}} \mathcal{L}^r, c_n \rangle] = \frac{1}{K} \sum_{k \in [K]} \mathcal{O}(\frac{\sigma_0^{0.5}}{M^2}) - \sum_{i \neq m_k} \Theta(\frac{1}{M}) \cdot \frac{1}{K} \sum_{k \in [K]} \mathcal{O}(\frac{\sigma_0^{0.5}}{M^3})$$

$$= \mathcal{O}(\frac{\sigma_0^{0.5}}{M^2}).$$

We can use the similar way to prove $\mathbb{E}[\langle \nabla_{\boldsymbol{\theta}^{(i)}} \mathcal{L}^r, \xi_l \rangle] = \mathcal{O}(\frac{\sigma_\xi}{M^2}), \forall l \in [L - 3]$.

For $\mathbb{E}[\langle \nabla_{\boldsymbol{\theta}^{(i)}} \mathcal{L}^r, v_n \rangle]$, it contains four cases: (1) $(\boldsymbol{X}^{(k)}, y^{(k)}) \in \Omega_n$ with $m_k = i$, (2) $(\boldsymbol{X}^{(k)}, y^{(k)}) \in \Omega_n$ with $m_k = j \neq i$, (3) $(\boldsymbol{X}^{(k)}, y^{(k)}) \in \Omega_{n',n}$ with $m_k = i$, and (4) $(\boldsymbol{X}^{(k)}, y^{(k)}) \in \Omega_{n',n}$ with $m_k = j \neq i$. Consequently, we calculate

$$
\mathbb{E}[\langle \nabla_{\boldsymbol{\theta}^{(i)}} \mathcal{L}^r, v_n \rangle] = \underbrace{\frac{1}{K} \sum_{\boldsymbol{X}^{(k)} \in \Omega_n} \mathbb{P}(m_k = i) y^{(k)} \cdot f \cdot \ell' \cdot \pi_{m_k} \cdot (1 - \pi_{m_k}) \cdot \sum_{l=1}^{L} (\boldsymbol{X}_l^{(k)})^\top \cdot (y^{(k)} v_n)}_{I_1}
$$

$$
+ \underbrace{\frac{1}{K} \sum_{\boldsymbol{X}^{(k)} \in \Omega_{n',n}} \mathbb{P}(m_k = i) y^{(k)} \cdot f \cdot \ell' \cdot \pi_{m_k} \cdot (1 - \pi_{m_k}) \cdot \sum_{l=1}^{L} (\boldsymbol{X}_l^{(k)})^\top \cdot (\epsilon^{(k)} v_n)}_{I_2}
$$

$$
+ \underbrace{\frac{1}{K} \sum_{\boldsymbol{X}^{(k)} \in \Omega_n} \mathbb{P}(m_k = j | j \neq i) y^{(k)} \cdot f \cdot \ell' \cdot \pi_{m_k} \cdot (-\pi_i) \cdot \sum_{l=1}^{L} (\boldsymbol{X}_l^{(k)})^\top \cdot (y^{(k)} v_n)}_{I_3}
$$

$$
+ \underbrace{\frac{1}{K} \sum_{\boldsymbol{X}^{(k)} \in \Omega_{n',n}} \mathbb{P}(m_k = j | j \neq i) y^{(k)} \cdot f \cdot \ell' \cdot \pi_{m_k} \cdot (-\pi_i) \cdot \sum_{l=1}^{L} (\boldsymbol{X}_l^{(k)})^\top \cdot (\epsilon^{(k)} v_n)}_{I_4}.
$$

Based on our analysis of $\mathbb{E}[\langle \nabla_{\boldsymbol{\theta}^{(i)}} \mathcal{L}^r, c_n \rangle]$ above, we calculate

$$
I_1 = \frac{1}{K} \sum_{\boldsymbol{X}^{(k)} \in \Omega_n} \Theta(\frac{1}{M}) \cdot \mathcal{O}(\sigma_0^{0.5}) \cdot \Theta(\frac{1}{M}) = \mathcal{O}(\frac{\sigma_0^{0.5}}{KM^2}) = \mathcal{O}(\frac{\sigma_0^{0.5}}{N^2 M^2}),
$$

where the last equality follows because $K = \Theta(N^2)$ derived in Lemma 10. Similarly, we calculate

$$
I_2 = \mathcal{O}(\frac{\sigma_0^{0.5}}{N^2 M^2}), \quad I_3 = \mathcal{O}(\frac{\sigma_0^{0.5}}{N^2 M^3}), \quad \text{and} \quad I_4 = \mathcal{O}(\frac{\sigma_0^{0.5}}{N^2 M^3}).
$$

Finally, we obtain

$$
\mathbb{E}[\langle \nabla_{\boldsymbol{\theta}^{(i)}} \mathcal{L}^r, v_n \rangle] = \mathcal{O}(\frac{\sigma_0^{0.5}}{N^2 M^2}).
$$

This completes the proof of Lemma 12. $\qquad \square$

**Lemma 13.** *At any epoch $t \in \{1, \cdots, T_1\}$, the following properties hold:*

$$
\mathbb{E}[\langle \nabla_{W^{(i)}} \mathcal{L}^e, v_n^* \rangle] - \mathbb{E}[\langle \nabla_{W^{(i)}} \mathcal{L}^e, v_n \rangle] = \mathcal{O}(\frac{N \sigma_0^{0.5}}{M}), \tag{15}
$$

$$
\mathbb{E}[\langle \nabla_{W^{(i)}} \mathcal{L}^e, c_n \rangle] = \mathcal{O}(\frac{N \sigma_0}{M}), \tag{16}
$$

$$
\mathbb{E}[\langle \nabla_{W^{(i)}} \mathcal{L}^e, \xi_l \rangle] = \mathcal{O}(\sigma_\xi), \forall l \in [L - 3]. \tag{17}
$$

*Proof.* We first prove Eq. (16). Based on Algorithm 1, FFN of each transformer $i$ will only be updated after the training data are sent to this expert. Hence, there are four cases for updating $\langle \nabla_{W^{(i)}} \mathcal{L}^e, c_n \rangle$:

(1) $\boldsymbol{X}^{(k)} \in \Omega_{+n, +n'}$.

(2) $\boldsymbol{X}^{(k)} \in \Omega_{+n, -n'}$.

(3) $\boldsymbol{X}^{(k)} \in \Omega_{-n, +n'}$.

(4) $\boldsymbol{X}^{(k)} \in \Omega_{-n, -n'}$.

Then we calculate

$$
\mathbb{E}[\langle \nabla_{W^{(i)}} \mathcal{L}^e, c_n \rangle] = \sum_{n \neq n'} \mathbb{P}(m_k = i) \Big( \mathbb{P}\{\boldsymbol{X}^{(k)} \in \Omega_{+n, +n'}\} \langle \nabla_{W^{(i)}} \mathcal{L}^e(\boldsymbol{X}_{+n, +n'}^{(k)}), c_n \rangle
$$

$$
+ \mathbb{P}\{\boldsymbol{X}^{(k)} \in \Omega_{+n, -n'}\} \langle \nabla_{W^{(i)}} \mathcal{L}^e(\boldsymbol{X}_{+n, -n'}^{(k)}), c_n \rangle
$$

$$+ \mathbb{P}\{\boldsymbol{X}^{(k)} \in \Omega_{-n,+n'}\}\langle \nabla_{W^{(i)}}\mathcal{L}^e(\boldsymbol{X}_{-n,+n'}), c_n\rangle$$

$$+ \mathbb{P}\{\boldsymbol{X}^{(k)} \in \Omega_{-n,-n'}\}\langle \nabla_{W^{(i)}}\mathcal{L}^e(\boldsymbol{X}_{-n,-n'}), c_n\rangle\Big).$$

Based on the expression of $\nabla_{W^{(i)}}\mathcal{L}^r$ in Eq. (13), we have

$$\langle \nabla_{W^{(i)}}\mathcal{L}^e(\boldsymbol{X}_{+n,+n'}), c_n\rangle = \ell' \cdot y_{+n,+n'} \cdot \Big(\boldsymbol{X}_{+n,+n'} \cdot \sum_{l=1}^{L} \text{softmax}(\boldsymbol{X}_{+n,+n'}^\top W_{KQ}(\boldsymbol{X}_{+n,+n'})_l)\Big)^\top c_n$$

$$= \ell' \cdot \Big(\boldsymbol{X}_{+n,+n'} \cdot \sum_{l=1}^{L} \text{softmax}(\boldsymbol{X}_{+n,+n'}^\top W_{KQ}(\boldsymbol{X}_{+n,+n'})_l)\Big)^\top c_n,$$

due to the fact that $y_{+n,+n'} = 1$. Similarly, we calculate

$$\langle \nabla_{W^{(i)}}\mathcal{L}^e(\boldsymbol{X}_{-n,-n'}), c_n\rangle = -\ell' \cdot \Big(\boldsymbol{X}_{+n,+n'} \cdot \sum_{l=1}^{L} \text{softmax}(\boldsymbol{X}_{-n,-n'}^\top W_{KQ}(\boldsymbol{X}_{-n,-n'})_l)\Big)^\top c_n.$$

Note that $c_n$ is orthogonal with $v_n$ and $v_{n'}$, such that $v_n^\top c_n = 0$. Additionally, we have $\xi_l^\top c_n = \mathcal{O}(\sigma_\xi)$. Furthermore, based on the fact $|\nu^\top W_{KQ}^{(i)}\mu| = \mathcal{O}(\sigma_0)$ at Stage I in Lemma 5, we have

$$\langle \nabla_{W^{(i)}}\mathcal{L}^e(\boldsymbol{X}_{+n,+n'}), c_n\rangle + \langle \nabla_{W^{(i)}}\mathcal{L}^e(\boldsymbol{X}_{-n,-n'}), c_n\rangle = \mathcal{O}(\sigma_0).$$

Similarly, we can prove

$$\langle \nabla_{W^{(i)}}\mathcal{L}^e(\boldsymbol{X}_{+n,-n'}), c_n\rangle + \langle \nabla_{W^{(i)}}\mathcal{L}^e(\boldsymbol{X}_{-n,+n'}), c_n\rangle = \mathcal{O}(\sigma_0).$$

Combining the above analysis, we obtain $\mathbb{E}[\langle \nabla_{W^{(i)}}\mathcal{L}^e, c_n\rangle] = \frac{N\sigma_0}{M}$, based on the fact that $\mathbb{P}(m_k = i) = \Theta(\frac{1}{M})$.

Differently from the four cases under the update of $\langle \nabla_{W^{(i)}}\mathcal{L}^e, c_n\rangle$, there exist another four cases for updating $\langle \nabla_{W^{(i)}}\mathcal{L}^e, v_n\rangle$:

(5) $\boldsymbol{X}^{(k)} \in \Omega_{+n',+n}$.

(6) $\boldsymbol{X}^{(k)} \in \Omega_{+n',-n}$.

(7) $\boldsymbol{X}^{(k)} \in \Omega_{-n',+n}$.

(8) $\boldsymbol{X}^{(k)} \in \Omega_{-n',-n}$.

We obtain

$$\mathbb{E}[\langle \nabla_{W^{(i)}}\mathcal{L}^e, v_n\rangle] = \mathbb{P}\{\boldsymbol{X}^{(k)} \in \Omega_{n,n'}\} \sum_{n' \neq n} \mathbb{P}(m_k = i)\langle \nabla_{W^{(i)}}\mathcal{L}^e(\boldsymbol{X}_{n,n'}^{(k)}), v_n\rangle$$

$$+ \mathbb{P}\{\boldsymbol{X}^{(k)} \in \Omega_{n',n}\} \sum_{n' \neq n} \mathbb{P}(m_k = i)\langle \nabla_{W^{(i)}}\mathcal{L}^e(\boldsymbol{X}_{n',n}^{(k)}), v_n\rangle.$$

We then prove Eq. (15). Note that by Lemma 5 and Lemma 6, for any $t \in [T_1]$, we have

$$|\text{softmax}(\boldsymbol{X}W_{KQ}^{(i)}\mu)_{l(\nu_1)} - \text{softmax}(\boldsymbol{X}W_{KQ}^{(i)}\mu)_{l(\nu_2)}| = \mathcal{O}(\sigma_0).$$

Additionally, by Lemma 11, we obtain $f(\boldsymbol{X}; \Theta, W^{(m)}, W_{KQ}^{(m)}) = \mathcal{O}(\sigma_0^{0.5})$. Consequently, at any epoch $t \in [T_1]$, we have

$$\mathbb{E}[\langle \nabla_{W^{(i)}}\mathcal{L}^e, v_n^*\rangle] - \mathbb{E}[\langle \nabla_{W^{(i)}}\mathcal{L}^e, v_n\rangle] \tag{18}$$

$$= \mathbb{P}\{\boldsymbol{X}^{(k)} \in \Omega_{n^*,n'}\} \sum_{n' \neq n^*} \mathbb{P}(m_k = i)\langle \nabla_{W^{(i)}}\mathcal{L}^e(\boldsymbol{X}_{n^*,n'}^{(k)}), v_{n^*}\rangle$$

$$+ \mathbb{P}\{\boldsymbol{X}^{(k)} \in \Omega_{n',n^*}\} \sum_{n' \neq n^*} \mathbb{P}(m_k = i)\langle \nabla_{W^{(i)}}\mathcal{L}^e(\boldsymbol{X}_{n',n^*}^{(k)}), v_{n^*}\rangle$$

$$- \mathbb{P}\{\boldsymbol{X}^{(k)} \in \Omega_{n,n''}\} \sum_{n'' \neq n} \mathbb{P}(m_k = i)\langle \nabla_{W^{(i)}}\mathcal{L}^e(\boldsymbol{X}_{n,n''}^{(k)}), v_n\rangle$$

$$- \mathbb{P}\{\boldsymbol{X}^{(k)} \in \Omega_{n'',n}\} \sum_{n'' \neq n} \mathbb{P}(m_k = i)\langle \nabla_{W^{(i)}}\mathcal{L}^e(\boldsymbol{X}_{n'',n}^{(k)}), v_n\rangle.$$

For any $\nabla_{W^{(i)}}\mathcal{L}^e(\boldsymbol{X}_{n^*,n'}^{(k)})$ and $\nabla_{W^{(i)}}\mathcal{L}^e(\boldsymbol{X}_{n,n''}^{(k)})$, based on the symmetric data points, we calculate

$$\langle \nabla_{W^{(i)}}\mathcal{L}^e(\boldsymbol{X}_{n^*,n'}), v_n \rangle - \langle \nabla_{W^{(i)}}\mathcal{L}^e(\boldsymbol{X}_{n^*,n'}), v_n \rangle$$

$$= \ell'(f(\boldsymbol{X}_{n^*,n'}^{(k)})) \cdot y_{n^*,n'} \cdot \boldsymbol{X}_{n^*,n'} \cdot \text{softmax}(\boldsymbol{X}_{n^*,n'}^\top W_{KQ}^{(i)} \boldsymbol{X}_{n^*,n'})$$

$$- \ell'(f(\boldsymbol{X}_{n,n''}^{(k)})) \cdot y_{n,n''} \cdot \boldsymbol{X}_{n,n''} \cdot \text{softmax}(\boldsymbol{X}_{n,n''}^\top W_{KQ}^{(i)} \boldsymbol{X}_{n,n''}).$$

Note that the function $\ell'$ is $1/4$-Lipschitz. Thus, by the mean value theorem,

$$|\ell(f(\boldsymbol{X}_{n^*,n'})) - \ell(f(\boldsymbol{X}_{n,n''}))| \leq \frac{1}{4}|f(\boldsymbol{X}_{n^*,n'}) - f(\boldsymbol{X}_{n,n''})| = \mathcal{O}(\sigma_0^{0.5}).$$

Additionally, the attention layer is frozen in Stage I. Hence, we obtain

$$\langle \nabla_{W^{(i)}}\mathcal{L}^e(\boldsymbol{X}_{n^*,n'}), v_n \rangle - \langle \nabla_{W^{(i)}}\mathcal{L}^e(\boldsymbol{X}_{n,n''}), v_n \rangle = \mathcal{O}(\sigma_0^{0.5}). \tag{19}$$

Putting Eq. (19) back to Eq. (18), we obtain

$$\mathbb{E}[\langle \nabla_{W^{(i)}}\mathcal{L}^e(\boldsymbol{X}_{n^*,n'}), v_n \rangle] - \mathbb{E}[\langle \nabla_{W^{(i)}}\mathcal{L}^e(\boldsymbol{X}_{n^*,n'}), v_n \rangle] = \mathcal{O}(\frac{N\sigma_0^{0.5}}{M}).$$

This completes the proof of Eq. (15).

To prove Eq. (17), we can similarly expand $\mathbb{E}[\langle \nabla_{W^{(i)}}\mathcal{L}^e, \xi_l \rangle]$ and apply $\xi_l = \mathcal{O}(\sigma_\xi)$, such that we skip the proof here. $\qquad\square$

According to Lemma 12, at $T_1 = \mathcal{O}(\eta^{-1}\sigma_0^{-0.5}M)$, we have

$$\mathbb{E}\left[(\theta_{T_1}^{(i)})^\top c_n\right] = (\theta_0^{(i)} - \eta_r T_1 \mathbb{E}[\nabla_{\boldsymbol{\theta}^{(i)}}\mathcal{L}^r])^\top c_n$$

$$= \mathcal{O}(\frac{1}{M}),$$

by letting $\eta_r = \Theta(\eta)$. At the same time, we obtain $\mathbb{E}\left[(\theta_{T_1}^{(i)})^\top v_n\right] = \mathcal{O}(\frac{1}{N^2 M})$.

Let $\boldsymbol{X}$ and $\tilde{\boldsymbol{X}}$ denote two data points, both with class signal $c_n$. Then we calculate

$$|h_i(\boldsymbol{X}; \boldsymbol{\Theta}) - h_i(\tilde{\boldsymbol{X}}; \boldsymbol{\Theta})| = \mathcal{O}(\frac{1}{N^2 M}).$$

Further, for any two data points $\boldsymbol{X}_1$ and $\boldsymbol{X}_2$ with different class signals $c_n$ and $c_{n'}$, we calculate

$$|h_i(\boldsymbol{X}_1; \boldsymbol{\Theta}) - h_i(\boldsymbol{X}_2; \boldsymbol{\Theta})| = \mathcal{O}(\frac{1}{M}) \ll |h_i(\boldsymbol{X}; \boldsymbol{\Theta}) - h_i(\tilde{\boldsymbol{X}}; \boldsymbol{\Theta})|.$$

Consequently, the router selects each expert $i \in \mathcal{M}_{n_i^*}$ for any data includes class signal $c_{n_i^*}$ with equal probability $\frac{1}{|\mathcal{M}_{n_i^*}|}$.

According to Eq. (15) in Lemma 13, there exists $T_1 = \mathcal{O}(\eta^{-1}\sigma_0^{0.5}M)$, such that we have

$$\mathbb{E}\left[\langle W_{T_1}^{(i)}, v_n^* \rangle - \langle W_{T_1}^{(i)}, v_n \rangle\right] = \mathcal{O}(\sigma_0) + \eta \cdot \left(\mathbb{E}[\langle \nabla_{W^{(i)}}\mathcal{L}^e, v_n^* \rangle] - \mathbb{E}[\langle \nabla_{W^{(i)}}\mathcal{L}^e, v_n \rangle]\right) \cdot T_1$$

$$= \Theta(\sigma_0^{0.5}).$$

This completes the proof of Proposition 1.

## E  PROOF OF PROPOSITION 2

Based on Proposition 1, the router will only send data within the same class $n_i^*$ to each expert $i \in \mathcal{M}_{n_i^*}$. Consequently, since Stage II, there is no need to focus on the gating network parameter.

**Lemma 14.** *At any epoch $t \in \{T_1 + 1, \cdots, T_2\}$, for any expert $i \in \mathcal{M}_n$, the following property holds:*

$$\mathbb{E}[\nu^\top \nabla_{W_{KQ}^{(i)}}\mathcal{L}^e \mu] = \begin{cases} \Omega(\frac{N\sigma_0^{0.5}}{M}), & \text{if } \mu = \nu = v_n, \\ \mathcal{O}(\sigma_0), & \text{otherwise.} \end{cases}$$

*Proof.* For $\mu = \nu = v_n$, we calculate

$$\mathbb{E}[v_n^\top \nabla_{W_{KQ}^{(i)}}\mathcal{L}^e v_n] = \sum_{k \in [K]} \mathbb{P}(\boldsymbol{X}^{(k)} \in \Omega_n) \cdot \mathbb{P}(m_k = i | \boldsymbol{X}^{(k)} \in \Omega_n) \tag{20}$$

$$\cdot \sum_{n' \neq n} \mathbb{P}(\boldsymbol{X}^{(k)} \in \Omega_{n,n'}) \cdot v_n^\top \nabla_{W_{KQ}^{(i)}}\mathcal{L}^e(\boldsymbol{X}_{n,n'}) v_n.$$

Based on the expression of $\nabla_{W_{KQ}^{(i)}} \mathcal{L}^e$ in Lemma 4, we calculate

$$v_n^\top \nabla_{W_{KQ}^{(i)}} \mathcal{L}^e(\boldsymbol{X}_{n,n'}) v_n$$

$$= v_n^\top \cdot y^{(k)} \cdot \ell' \cdot \sum_{l=1}^{L} \boldsymbol{X}^{(k)} [\mathrm{diag}\,(\boldsymbol{p}_l^{(i)}) - \boldsymbol{p}_l^{(m_k)} (\boldsymbol{p}_l^{(i)})^\top] (\boldsymbol{X}_{n,n'})^\top W^{(i)} (\boldsymbol{X}_{n,n'})_l^\top \cdot v_n$$

$$= \Omega(\sigma_0^{0.5}),$$

which follows from the fact that $\ell' = \Theta(1)$, $\boldsymbol{p}_l^{(i)} = \Theta(1)$, and $(W^{(i)})^\top v_n = \Omega(\sigma_0^{0.5})$ in Phase II. Additionally, we have

$$\sum_{k \in [K]} \mathbb{P}(\boldsymbol{X}^{(k)} \in \Omega_n) \cdot \mathbb{P}(m_k = i | \boldsymbol{X}^{(k)} \in \Omega_n) = K \cdot \frac{1}{N} \cdot \frac{1}{|\mathcal{M}_n|} = \Theta\left(\frac{N}{M}\right),$$

which follows from the fact that $K = \Theta(N^2)$ derived in Lemma 10 and $\frac{1}{|\mathcal{M}_n|} = \Theta(\frac{1}{M})$ derived in Lemma 8.

Based on the above analysis, we obtain $\mathbb{E}[v_n^\top \nabla_{W_{KQ}^{(i)}} \mathcal{L}^e v_n] = \Omega(\frac{N \sigma_0^{0.5}}{M})$ in Eq. (20).

Next, we take an example of $\nu = c_n$ and $\mu = c_n$ to prove $\mathbb{E}[\nu^\top \nabla_{W_{KQ}^{(i)}} \mathcal{L}^e \mu] = \mathcal{O}(\sigma_0)$. Other cases can be similarly proved. We calculate

$$\mathbb{E}[c_n^\top \nabla_{W_{KQ}^{(i)}} \mathcal{L}^e c_n] = \sum_{k \in [K]} \mathbb{P}(\boldsymbol{X}^{(k)} \in \Omega_n) \cdot \mathbb{P}(m_k = i | \boldsymbol{X}^{(k)} \in \Omega_n)$$

$$\cdot \sum_{n' \neq n} \mathbb{P}(\boldsymbol{X}^{(k)} \in \Omega_{n,n'}) \cdot c_n^\top \nabla_{W_{KQ}^{(i)}} \mathcal{L}^e(\boldsymbol{X}_{n,n'}) c_n.$$

Given $(W^{(i)})^\top c_n = \mathcal{O}(\sigma_0)$ derived in Proposition 1, we can obtain $\mathbb{E}[\nu^\top \nabla_{W_{KQ}^{(i)}} \mathcal{L}^e \mu] = \mathcal{O}(\sigma_0)$. $\qquad \square$

Based on Lemma 14, only if both $\mu$ and $\nu$ equal the classification $v_n$, the attention weight $\nu^\top \nabla_{W_{KQ}^{(i)}} \mathcal{L}^e \mu$ will significantly increase. Then there exists epoch $T_2 = T_1 + \mathcal{O}(\eta_a^{-1} \sigma_0^{-0.5} N^{-1} M)$, such that

$$\mathbb{E}[v_n^\top W_{KQ}^{(i)} v_n] = \mathcal{O}(\sigma_0) + (T_2 - T_1) \cdot \mathbb{E}[v_n^\top \nabla_{W_{KQ}^{(i)}} \mathcal{L}^e v_n]$$

$$= \Theta(1).$$

Based on Lemma 6, at epoch $T_2$, we obtain that the corresponding attention score satisfies

$$p_{\mu=v_n,\nu=v_n}^{(i)}(\boldsymbol{X}(n)) = \Theta(1).$$

While for the other pairs, their attention scores remain at $\mathcal{O}(\sigma_0)$.

# F    PROOF OF THEOREM 1

Finally, in Stage III, we fix the attention layer again and fine-tune FFNs. Due to the transformer specialization in Stage I and the specialization strengthened in Stage II, the FFNs only focus on increasing $\langle W^{(i)}, v_n \rangle$ in Stage III, where $i \in \mathcal{M}_n$. In the following, we exploit the strong convexity of the loss function to prove Theorem 1.

**Lemma 15.** *For $\ell(z) = \log(1 + \exp(-z))$, as $z \to +\infty$, we have $\ell(z) = \mathcal{O}(\exp(-z))$.*

*Proof.* As $z \to +\infty$, $e^{-z} \to 0$. By Taylor expansion,

$$\log(1 + e^{-z}) = e^{-z} - \frac{1}{2} e^{-2z} + O(e^{-3z}) < e^z,$$

i.e., $\log(1 + e^{-z}) = \mathcal{O}(e^{-z})$ as $z \to +\infty$. $\qquad \square$

**Lemma 16.** *For any $t \geq T_2$, the following property hold:*

$$|\langle \nabla_{W^{(i)}} \mathcal{L}^e, v_n \rangle| = \mathcal{O}\left(\frac{1}{1 + \sigma_0^{0.5}}\right).$$

*Proof.* Based on Lemma 11, we have $f(\boldsymbol{X}; \boldsymbol{\Theta}, W^{(i)}, W_{KQ}^{(i)}) = \Omega(\sigma_0^{0.5})$ after the completion of Stage I. Consequently, given $\ell'(z) = \frac{-1}{1+\exp(z)}$, we have

$$|\ell'(z)| = \mathcal{O}(\frac{1}{1 + \exp{(\sigma_0^{0.5})}}) \leq \mathcal{O}(\frac{1}{1 + (\sigma_0)^{0.5}}),$$

where the last inequality is by Taylor expansion. Then substituting $|\ell'(z)| = \mathcal{O}(\frac{1}{1+(\sigma_0^{0.5})})$ into the expression of $\nabla_{W^{(i)}} \mathcal{L}^e$ in Eq. (13) completes the proof. $\square$

Given $T^* = T_2 + \mathcal{O}(\log(\epsilon^{-1}))$ and $|\langle \nabla_{W^{(i)}} \mathcal{L}^e, v_n \rangle| = \mathcal{O}(\frac{1}{1+\sigma_0^{0.5}})$ in Lemma 16, we have

$$|(W^{(i)})^\top v_n| \geq \mathcal{O}(\sigma_0^{0.5}) + \mathcal{O}(\log(\epsilon^{-1})) \cdot |\langle \nabla_{W^{(i)}} \mathcal{L}^e, v_n \rangle|$$
$$= \mathcal{O}(\log(\epsilon^{-1})).$$

Based on Proposition 2, we further obtain $|f(\boldsymbol{X}; \boldsymbol{\Theta}, W^{(i)}, W_{KQ}^{(i)})| \geq \mathcal{O}(\log(\epsilon^{-1}))$.

Based on Lemma 15, we obtain

$$\mathcal{L}^e(\boldsymbol{X}) = \frac{1}{K} \sum_{k \in [K]} \ell(y^{(k)} \cdot f(\boldsymbol{X}; \boldsymbol{\Theta}, W^{(i)}, W_{KQ}^{(i)}))$$

$$= \frac{1}{K} \sum_{k \in [K]} \mathcal{O}\Big( \exp{(-y^{(k)} \cdot f(\boldsymbol{X}; \boldsymbol{\Theta}, W^{(i)}, W_{KQ}^{(i)}))}\Big)$$

$$= \mathcal{O}(\epsilon),$$

which completes the proof of Theorem 1.

## G  PROOF OF LEMMA 1

In the MoE model without expert-specific attention, the routing strategy up to the end of Stage I is identical to that in Proposition 1, since the attention layers are fixed (not trained) in this setting. Thus, after Stage I, the gating network and experts have converged such that each expert specializes in a class, as described in Proposition 1.

The output of expert $i$ for an input $\boldsymbol{X}$ is given by

$$f(\boldsymbol{X}; \boldsymbol{\Theta}, W^{(i)}) = \sum_{l=1}^{L} (W^{(i)})^\top \boldsymbol{X}_l.$$

Recall that in the data model (Definition 1), each input $\boldsymbol{X}$ consists of three signals and $L - 3$ Gaussian noise vectors. Consequently, we can decompose the model output as:

$$f(\boldsymbol{X}; \boldsymbol{\Theta}, W^{(i)}) = \sum_{\mu \in \mathcal{C} \cup \mathcal{U}} (W^{(i)})^\top \mu + \sum_{j=1}^{L-3} W^{(i)} \xi_j,$$

where the "signal terms" involve fixed vectors $(c_n, v_n, v_{n'})$ and the rest is the summation of the projections of the Gaussian noise tokens.

Upon the completion of Stage I, by Proposition 1, $W^{(i)}$ aligns with $v_n$ and can suppress the distractor to some extent. However, it cannot fully remove the interference of noise tokens, because the model lacks an attention mechanism to select the relevant tokens. Consequently, for any $t \geq T_1 + 1$, we have

$$(W^{(i)})^\top v_n = \mathcal{O}(\sigma_0^{0.5}) - \eta \cdot (t - T_1) \cdot \langle \nabla_{W^{(i)}} \mathcal{L}^e, v_n \rangle,$$
$$(W^{(i)})^\top \xi_j = \mathcal{O}(\sigma_\xi) - \eta \cdot (t - T_1) \cdot \langle \nabla_{W^{(i)}} \mathcal{L}^e, \xi_j \rangle.$$

Hence, after $t - T_1 = \Theta(\log(\epsilon^{-1}))$ steps,

$$f(\boldsymbol{X}; \boldsymbol{\Theta}, W^{(i)}) = \Theta\left( (1 - L\sigma_\xi) \log(\epsilon^{-1}) \right),$$

where the $(1 - L\sigma_\xi)$ factor captures the reduction in effective margin due to cumulative noise.

Now, substituting this into the logistic loss, for small $\epsilon$, we have:

$$\mathbb{E}[\mathcal{L}^e(\boldsymbol{X})] = \log\left( 1 + \exp(-y \cdot f(\boldsymbol{X}; \boldsymbol{\Theta}, W^{(i)})) \right) = \Theta\Big( \exp\left( -y \cdot f(\boldsymbol{X}; \boldsymbol{\Theta}, W^{(i)}) \right) \Big)$$

$$\geq \Theta(\epsilon^{1-L\sigma_\xi}).$$

This completes the proof of Lemma 1.

