# OpenReview forum: "Mixture-of-Transformers Learn Faster: A Theoretical Study on Classification Problems"
_ICLR.cc/2026/Conference — Submitted to ICLR 2026_

### Official Review · Reviewer_dCo3 · 2025-10-29

**Soundness:** 3
**Presentation:** 3
**Contribution:** 1
**Rating:** 2
**Confidence:** 5

**Summary:**

This paper provides the first theoretical convergence and generalization analysis of the Mixture-of-Transformers (MoT) architecture, where, unlike the canonical Mixture-of-Experts (MoE), each expert contains a separate dedicated attention block, i.e., instead of being an FFN, each expert is a transformer block (attention head followed by an FFN). Based on a single MoT layer and assumptions on the input data, the paper theoretically shows that MoT converges faster than a two-headed one-layer transformer model, and achieves better performance than an attention-absent MoE model.

**Strengths:**

1. The first theoretical training dynamic analysis of the MoT architecture, providing insights into the different phases of training of the architecture.

2. The paper is well-structured and easy to follow.

**Weaknesses:**

1. **Lack of novelty:** The theoretical analysis closely follows the setup in [1], e.g., the data model and the network model are very similar. Therefore, the characterization of training dynamics over multiple training phases of MoE (e.g., router exploration phase, expert specialization phase, etc.) does not provide any significant new insights. The only contribution can be the characterization of the role of the attention layer in the MoE transformer block.

2. **Inappropriate baselines:** The paper compares the proposed architecture with attention-absent MoE and the two-headed transformer. However, usually an MoE transformer block includes a shared attention (or multi-head attention) layer followed by FFN experts. Moreover, the theoretical advantage of MoE over dense models has already been established in [1]. Therefore, the paper should demonstrate the theoretical advantage of MoT over shared attention followed by FFN expert based setup, and also should verify this in experiments.

3. **Unfair comparison:** The comparison with the two baselines (two-headed transformer and attention-absent MoE) is unfair. In the vanilla transformer case, there are only two attention heads, whereas the number of expert transformer blocks in the proposed architecture is large ($\Omega(NlogN)$). It has not been theoretically established whether the advantage of MoT remains when the number of attention heads in the vanilla transformer increases from two. For the case of attention-absent MoE, the number of neurons in each FFN expert is $1$ (i.e., $W^{(i)}=\mathbb{R}^{d\times1}$). It has not been theoretically established whether the advantage of MoT remains when the number of neurons in the attention-absent MoE increases from one.

[1] Zixiang Chen, Yihe Deng, Yue Wu, Quanquan Gu, and Yuanzhi Li. Towards understanding the mixture-of-experts layer in deep learning. Advances in neural information processing systems, 35: 23049–23062, 2022

**Questions:**

In stage I, the authors claim that the experts become specialized. However, in this stage, the routers are still exploring. Can you elaborate, why the experts become specialized despite the exploration of the routers?

---

> ### Author Response · Authors · 2025-11-26
>
> Thank you for your thorough reviews and constructive comments. We provide our responses to your comments below and have made major revisions in our revised manuscript. To enhance clarity, we have highlighted the revised text in blue for easy identification.
>
> **Q1**. Lack of novelty: The theoretical analysis closely follows the setup in [1], e.g., the data model and the network model are very similar. Therefore, the characterization of training dynamics over multiple training phases of MoE (e.g., router exploration phase, expert specialization phase, etc.) does not provide any significant new insights. The only contribution can be the characterization of the role of the attention layer in the MoE transformer block.
>
> **A1**:Thank you for this comment. We would like to clarify that, although our data model is inspired by [1], the theoretical questions, architectural setting, and resulting conclusions are fundamentally different, and our analysis provides several insights that are not present in [1].
>
> - **Different architecture and different theoretical goals**: [1]) study an attention-absent MoE layer with simple linear experts. In contrast, our work analyzes Mixture-of-Transformers (MoT), where each expert contains both a self-attention module and an FFN. The resulting training dynamics fundamentally depend on the interaction between attention alignment and expert specialization, which is entirely absent in [1]. This richer architecture leads to several behaviors that cannot be captured by the model in [1], including attention alignment (Proposition 2), specialization-driven acceleration of convergence (Theorem 1), and noise suppression via attention (Lemma 1).
>
> - **Our contributions go far beyond the role of attention**: A key novelty is that we explicitly characterize all three stages of MoT training (i.e., FFN specialization, attention alignment, and FFN fine-tuning with fast convergence) under a unified analysis. These findings do not follow from [1], which does not involve attention, does not analyze the attention–router coupling, and does not provide $\epsilon$-accurate convergence guarantees under Gaussian noise.
>
> - We prove a result that clearly shows **the benefit of MoT compared to attention-absent MoE**: Using our analysis technique, we show that the attention-absent MoE cannot reduce the loss below any fixed $\epsilon>0$ due to the inevitable Gaussian distractor noise. In contrast, MoT can reduce the error to arbitrarily small $\epsilon$ because the attention layer aligns to the class-specific direction and suppresses noise. This clear separation result is entirely new and does not appear in [1].
>
> - Our **proof techniques** are substantially different from those in [1]: While [1] relies primarily on norm growth and router-update stability arguments, our proof uses several new tools, including: (a) conflicting-gradient analysis: showing how attention alignment reduces gradient interference and induces stage transitions; (b) feature concentration under attention weights: proving that attention suppresses noise in a way that cannot be achieved by FFN-only experts; (c) curvature-dependent convergence: deriving a geometric rate after attention alignment, which does not follow from the techniques in [1]. These techniques are required to handle the self-attention dynamics and are not extensions of Chen et al.’s arguments.
> In summary, although the data model has a similar form, which is standard in theoretical MoE analyses, the architectures, questions, insights, and proof methods differ substantially from [1]. Our work provides new theoretical guarantees and conceptual understanding that are not addressed by prior MoE studies.

---

> ### Author Response · Authors · 2025-11-26
>
> **Q2**. Inappropriate baselines.
>
> **A2**: Thank you for pointing this out. We would like to clarify the choice of baselines and how they relate to the theoretical claims of our paper.
>
> (1) Our primary baseline is the **multi-head transformer**, not a two-headed transformer.
> This baseline represents the standard dense transformer architecture with shared attention and shared FFN layers. By comparing MoT against this model, we directly evaluate the benefits of expert specialization and attention alignment, which are central to our theoretical analysis. The performance gains in Figure 2 (and the additional results in the Appendix) demonstrate that MoT achieves faster convergence and lower error once specialization emerges, even when compared to a transformer with full parameter sharing.
>
> (2) The comparison to **attention-absent MoE** is intentional, as it isolates the role of attention. While we agree that standard MoE transformers normally include a shared attention layer followed by FFN experts, the attention-absent MoE baseline serves a very different purpose in our analysis. It allows us to test, in a clean and controlled way, the function of specialized attention within MoT. This baseline directly reflects the theoretical separation result we establish: attention-absent MoE cannot reduce the loss below a constant $\epsilon$ due to Gaussian distractor noise. While MoT can suppress this noise via attention alignment and reduce the loss to arbitrarily small $\epsilon$. This provides the first step toward understanding how specialized attention fundamentally changes the dynamics of expert learning, which cannot be observed with baselines that share the same attention.
>
> In summary, our baseline choices reflect two complementary goals: (1) The comparison with dense transformer baseline evaluates the benefit of specialization and class-dependent attention alignment. (2) The comparison with attention-absent MoE baseline isolates and highlights the indispensable contribution of attention to suppressing noise and improving convergence. We have clarified these motivations in the revised manuscript.
>
> **Q3**. Unfair comparison.
>
> **A3**: Thank you for the question. We first clarify that **our baseline is a multi-head transformer, not a two-head transformer**. In our experiments, we set the number of attention heads in the baseline to be exactly the same as the number of experts (heads) in MoT, ensuring a fair comparison. The theoretical convergence rate of $O(\epsilon^{-1})$ for multi-head transformers follows from existing analyses (e.g., Yang et al., 2025), which study a two-head transformer **only because they focus on a two-mixture classification problem**. In our K-mixture setting, adding more heads (from 2 to K) does not improve the convergence rate; we observe empirically that multi-head transformers with K heads still retain the same $O(\epsilon^{-1})$ behavior. Therefore, the advantage of MoT over the dense transformer baseline does not stem from differences in head count, but from expert specialization and class-dependent attention alignment, consistent with our theoretical results.
>
> Regarding the attention-absent linear MoE baseline, our goal is to cleanly isolate and characterize the benefit of attention alignment, independent of the nonlinear capacity of individual experts. Studying MoT with linear FFNs allows us to prove that attention alignment suppresses Gaussian distractor noise and enables MoT to reduce the loss to arbitrarily small $\epsilon$, while attention-absent MoE cannot. This provides a fundamental, first-step theoretical insight into the role of specialized attention. In contrast, our experiments use realistic, nonlinear neural networks as FFN experts. Empirically, we observe that as the data distribution becomes more complex, the performance gap between MoT and the attention-absent MoE becomes larger. This precisely matches the theoretical intuition that attention becomes increasingly valuable when distractor noise or class entanglement grows. Additionally, this demonstrates that the insights derived from the simplified theoretical model transfer meaningfully to practical settings.
> Although [1] includes ReLU in their theoretical FFN definition, their real-data experiments still require much more complex expert networks than those analyzed in their theory. This reflects a common pattern in theoretical deep learning: starting from simplified models to extract general principles, then validating those principles empirically in richer architectures. Our approach follows this standard methodology.

---

> ### Author Response · Authors · 2025-11-26
>
> **Q4**. In stage I, the authors claim that the experts become specialized. However, in this stage, the routers are still exploring. Can you elaborate, why the experts become specialized despite the exploration of the routers?
>
> **A4**: Thank you for the question. We would like to clarify that our Stage I is fundamentally different from the two-stage progression analyzed in [1]. In [1], the authors identify a very short “exploration stage’’ of length $T_1^{[1]}=\lfloor \eta^{-1}\sigma_0^{0.5}\rfloor$, which is independent of the number of experts $M$, followed by a router-learning stage of length $T_2^{[1]}=\lfloor \eta^{-1}M^{-2}\rfloor$, which decreases as the number of experts increases. These time scales are somewhat counter-intuitive: as $M$ grows, one would expect the router to require more exploration, not less. Consequently, the specialization of experts and the stabilization of the router happen in two separate stages, and neither of these stages scales naturally with $M$.
>
> In contrast, our analysis shows that both router learning and expert specialization evolve within a single Stage-I window governed by $T_1=O(\eta^{-1}\sigma_0^{-0.5}M)$, which scales linearly with the number of experts and therefore aligns with intuition. Importantly, by the end of $T_1$, we prove that (i) **the router has already stabilized on a partition of the class space**, and (ii) **each FFN expert has specialized to its corresponding region (Proposition 1)**. Thus, in our framework, specialization is not delayed to a later stage; it naturally emerges alongside router exploration as part of the same training dynamics.
>
> This unified characterization of specialization and router stabilization is not present in [1] and represents another technical contribution of our work.
>
> **We hope our responses have addressed your questions. If so, we would be sincerely grateful if you would consider raising your rating to reflect this. Of course, we are happy to answer any further questions you may have. Thank you again for the time and thoughtful attention you have given to our work.**

---

### Official Review · Reviewer_Ru49 · 2025-10-30

**Soundness:** 2
**Presentation:** 3
**Contribution:** 2
**Rating:** 4
**Confidence:** 2

**Summary:**

The authors propose a theoretical framework for analyzing "Mixture-of-Transformers" (MoT), an architecture where each expert is a full transformer block (both self-attention and FFN) rather than just an FFN layer, as in traditional MoE. The paper's primary contribution is a theoretical analysis of MoT's convergence properties on a synthesized N-mixture classification task. To facilitate this analysis, the authors introduce a specific three-stage training algorithm that involves sequentially freezing and training the FFN and attention components. Under this setup, they prove that MoT achieves an $\mathcal{O}(\log(\epsilon^{-1}))$ convergence rate to an $\epsilon$-accurate solution. This is exponentially faster than the $\mathcal{O}(\epsilon^{-1})$ rate established for standard (non-gated) transformers on similar mixture problems. The authors argue this speed-up stems from expert specialization, which decomposes the complex, non-convex problem into a set of simpler, strongly convex sub-problems. They provide experiments on modified CIFAR and NLP datasets to validate these findings.

**Strengths:**

- **Clean theoretical result (in isolation):** The $\mathcal{O}(\log(\epsilon^{-1}))$ vs. $\mathcal{O}(\epsilon^{-1})$ contrast is stark and theoretically elegant. Proving that the attention-absent MoE has a fundamental error floor (Lemma 1) is a good theoretical contribution, highlighting why attention specialization is necessary in this problem setup.

- **clarity:** The paper is well-organized and clearly written. It systematically presents the model, the training algorithm, and the theoretical analysis, making the core ideas accessible.
- **Insightful analysis:** The analysis of the three-stage training process provides valuable insights into the distinct roles of the FFN and attention layers in specialization. Proposition 1 (FFN specialization) and Proposition 2 (attention training) build a clear narrative for how the model learns.

**Weaknesses:**

- **(Maybe) overly simplified theoretical model:** As detailed under "Soundness," the theoretical analysis rests on a foundation of strong assumptions (orthogonal data, single-layer model, merged matrices) that may not hold in practice. This severely limits the direct applicability of the findings to the deep, complex transformers used in the real world. The theory does not account for the interactions between multiple layers of experts.

- **lack of realistic empirical validation:** The experiments are conducted on artificially modified datasets to fit the theory. The absence of results on standard, unmodified benchmarks is a major weakness and makes it impossible to judge the practical utility of the model and the theory. Furthermore, the paper does not compare its staged training algorithm against standard end-to-end training, which is a critical missing baseline.

**Questions:**

- Could you please provide the experimental results (error curves analogous to Figure 2) on the unmodified, standard CIFAR-10 and CIFAR-100 datasets?

- Have you performed experiments training the MoT model end-to-end, rather than with the proposed three-stage algorithm? How do the convergence speed and final performance compare?

- The theoretical analysis relies heavily on assumptions like data orthogonality and a single-layer architecture. Could you elaborate on the potential challenges in extending your analysis to settings with correlated features and deep, multi-layer MoT models? Which of your assumptions do you believe are most critical to the final result, and which might be relaxed?

---

> ### Author Response · Authors · 2025-11-26
>
> Thank you for your thorough reviews and constructive comments. We provide our responses to your comments below and have made major revisions in our revised manuscript. To enhance clarity, we have highlighted the revised text in blue for easy identification.
>
> **Q1**. (Maybe) overly simplified theoretical model: As detailed under "Soundness," the theoretical analysis rests on a foundation of strong assumptions (orthogonal data, single-layer model, merged matrices) that may not hold in practice. This severely limits the direct applicability of the findings to the deep, complex transformers used in the real world. The theory does not account for the interactions between multiple layers of experts.
>
> **A1**:  Thank you for raising this important point. We emphasize that simplified assumptions of this type are standard and often necessary in theoretical studies of transformers and MoE systems. The goal of such models is to isolate the essential mechanisms that govern learning dynamics rather than to reproduce every detail of practical systems. Controlled assumptions such as orthogonal data and single layer architectures make rigorous and interpretable guarantees possible and follow a well established tradition in the theory community. Our experiments demonstrate that the qualitative insights obtained under these assumptions remain robust when evaluated on full architectures and real world datasets. We address each aspect below.
>
> (1) The orthogonal data model follows a standard and widely used paradigm in theoretical studies of transformers and sparse MoE systems. This setting has been adopted in Chen et al. (2022), Huang et al. (2024), Yang et al. (2025), and Li et al. (2025) to derive precise characterizations of representation learning, attention alignment, and expert specialization. Although orthogonality is a simplification, it retains the essential elements that drive the behavior of MoT, such as distractor signals, feature selection, and gradient conflict. Our empirical evaluations on CIFAR ten and CIFAR one hundred, Amazon Polarity, and YouTube Comments show that the qualitative behaviors predicted by the theory, including specialization and the acceleration of convergence after alignment, remain consistent on real data. Extending the analysis to correlated data is an interesting and technically demanding direction.
>
> (2) We analyze a one layer transformer expert because this setting enables a tractable study of how attention alignment, expert specialization, and router stabilization interact during training. This abstraction is consistent with the theoretical transformer literature and provides a first step toward understanding MoT architectures. Even in this simplified model, the coupling among attention, FFNs, and the gating network presents significant analytic challenges. Our experiments with multi layer vision transformers confirm that the mechanisms identified by the theory, including specialization, noise suppression through attention, and improved convergence, continue to hold in practice.
>
> (3) We merge the value, key, and query matrices into simplified linear operators to keep the analysis tractable. This follows common practice in theoretical studies of transformers, including Huang et al. (2024a), Huang et al. (2024b), and Zhang et al. (2024), where these matrices are combined when their separate roles do not affect the essential mechanisms being analyzed. The merged representation still captures the effect of attention alignment, which is central to the MoT convergence guarantees. Empirically, full attention parameterization yields the same qualitative behavior, indicating that this simplification does not compromise the insights.
>
> (4) We agree that analyzing multiple layers of experts is an important and open question. Doing so requires understanding how specialization and attention alignment propagate across layers and how router behavior evolves when deeper interactions arise. These phenomena introduce mathematical challenges that exceed the scope of current theoretical tools for MoE or transformer systems. Our work provides the first theoretical characterization of MoT training dynamics in any setting and offers a foundation for future analyses of deeper architectures. Our experiments with multi layer transformers show that the central insights of the theory continue to hold in practice.

---

> ### Author Response · Authors · 2025-11-26
>
> **Q2**. lack of realistic empirical validation: The experiments are conducted on artificially modified datasets to fit the theory. The absence of results on standard, unmodified benchmarks is a major weakness and makes it impossible to judge the practical utility of the model and the theory. Could you please provide the experimental results (error curves analogous to Figure 2) on the unmodified, standard CIFAR-10 and CIFAR-100 datasets?
>
> **A2**: Thank you for the helpful suggestion. We clarify that our modifications to CIFAR 10 and CIFAR 100 do not simplify the datasets. They add distractor pixels that introduce structured noise consistent with our data model in Definition 1, making the task more challenging and allowing us to directly test the theory’s predictions on noise suppression and attention alignment. We also include experiments on standard, unmodified datasets such as Amazon Polarity and YouTube Comments, which show that the insights from the theory transfer to real world data.
>
> In response to your request, we have added results on the original CIFAR 10 and CIFAR 100 datasets. The error curves are provided in **Figure 7 and Figure 8** of the appendix. These results demonstrate that the core behaviors predicted by our analysis, including expert specialization, attention driven noise reduction, and faster convergence after alignment, remain evident on the unmodified datasets.
>
> **Q3**. Furthermore, the paper does not compare its staged training algorithm against standard end-to-end training, which is a critical missing baseline. Have you performed experiments training the MoT model end-to-end, rather than with the proposed three-stage algorithm? How do the convergence speed and final performance compare?
>
> **A3**: Thank you for raising this point. We have added experiments comparing end to end training with our three stage training dynamics, reported in Fig. 5 and Fig. 8. When the MoT model is trained fully end to end, the convergence rate matches that of the multi head transformer baseline rather than the improved behavior achieved under our staged training. This occurs because, in the early iterations, all experts receive nearly identical gradients, so no meaningful specialization develops and the router continues to route inputs almost randomly, as confirmed by the routing histories in our added Fig. 6.
>
> In contrast, the three stage view separates the periods in which experts differentiate, attention aligns with the class specific direction, and the model enters the fast convergence regime. This decomposition is not intended as a practical training algorithm, but as a conceptual tool that reveals why specialization emerges and why convergence accelerates once alignment occurs.

---

> ### Author Response · Authors · 2025-11-26
>
> **Q4**. The theoretical analysis relies heavily on assumptions like data orthogonality and a single-layer architecture. Could you elaborate on the potential challenges in extending your analysis to settings with correlated features and deep, multi-layer MoT models? Which of your assumptions do you believe are most critical to the final result, and which might be relaxed?
>
> **A4**: Thank you for the thoughtful question.  We refer the reviewer to our response A1 for a detailed justification of the assumptions adopted in the theoretical model. Here we focus on clarifying the challenges involved in extending the analysis beyond these assumptions and the extent to which they are necessary for our results.
>
> (1) Relaxing the **orthogonality assumption** introduces several technical difficulties. In the orthogonal setting, gradients from different components of the data can be separated clearly, which allows us to quantify conflicting gradients responsible for expert specialization and attention alignment. When features are correlated, these gradients interact in a more complex manner, and the error decomposition used in our proofs no longer separates cleanly. Router updates also become harder to characterize because routing probabilities depend on mixtures of correlated signals. Extending the theory to correlated data would therefore require new tools for analyzing how attention filters correlated noise and how expert specialization emerges when gradients are no longer orthogonal. We believe the assumption of orthogonality is mainly a technical device that helps isolate the core mechanisms. Empirically, our experiments on real world datasets show that the qualitative behaviors predicted by the theory remain valid even when features are not orthogonal.
>
> (2) Extending the analysis to **deep MoT architectures** would require tracking how specialization and attention alignment propagate across layers. In a multi layer setting, experts in upper layers depend on the representations produced by earlier layers, and the router must learn to route tokens based on features that are themselves evolving through the depth of the model. This introduces recursive dependencies between alignment in one layer and the specialization of subsequent layers. Quantifying these dependencies is significantly more difficult because errors and gradient signals accumulate across layers. The single layer assumption is therefore important for analytical tractability, and relaxing it would require new techniques for controlling representation drift and cross layer interactions. Although challenging, this direction is promising and we view the present work as a first step in this broader research agenda.
>
> Among the assumptions we use, the single layer architecture is the most critical **for obtaining closed form convergence guarantees**. The orthogonality assumption is helpful for separating gradient components but is not fundamental to the underlying mechanism, and we expect it can be relaxed with more advanced proof techniques. The assumption of merged attention matrices is also not essential, because our empirical results using full attention parameterization confirm that the theoretical insights remain intact. We believe that the core mechanisms identified by the theory, including expert specialization, attention alignment, and the resulting improvement in convergence, should generalize to deep models and correlated data distributions. Proving these generalizations formally will require substantially different analytic tools.
>
> **We hope our responses have addressed your questions. If so, we would be sincerely grateful if you would consider raising your rating to reflect this. Of course, we are happy to answer any further questions you may have. Thank you again for the time and thoughtful attention you have given to our work.**

---

### Official Review · Reviewer_dwgC · 2025-10-31

**Soundness:** 3
**Presentation:** 3
**Contribution:** 3
**Rating:** 6
**Confidence:** 3

**Summary:**

This paper presents the first unified theoretical framework for Mixture-of-Transformers (MoT), where each transformer block acts as a full expert with its own attention and feed-forward layers, governed by a continuously trained gating network. The authors develop a three-stage training algorithm and prove that MoT achieves near-zero prediction loss in O(log⁡(ϵ^{−1})) steps—substantially faster than the O(ϵ^{−1}) rate for standard transformers. Experiments on image classification benchmarks validate the theoretical findings, showing that MoT outperforms both multi-head transformers and attention-absent MoE models, especially on complex tasks. The work provides practical guidance for designing efficient large-scale models and highlights the critical role of attention specialization in expert architectures.

**Strengths:**

(1) Provides the first rigorous analysis of full-transformer specialization, bridging a major gap in the MoE literature. The three-stage training procedure is interesting and isolates the roles of FFN and attention specialization.
(2) Proves faster convergence rates for MoT compared to standard baselines, supported by detailed proofs.
(3) Experiments on CIFAR-10/100 datasets robustly support the theoretical claims, demonstrating practical impact.

**Weaknesses:**

(1) For MoE, we typically have a one-stage training that learn all the model parameters (Attn, MLP, Gating) altogether. This paper proposed 3-stage training, which will complicate the training procedure. It might be better to discuss some ablation analysis, such as how the analysis will change for different stages of training, e.g., 2-stage or one single stage.

(2) While experiments look solid, they focus on relatively small-scale model (e.g., lightweight Vision Transformer) and benchmarks. Results on larger model and larger datasets would strengthen the claims, such as ImageNet. Also, it might be better to include some popular language benchmarks for transformer-based language models, such as GLUE, MMLU, HellaSwag, PIQA, etc.

(3) According to some studies on Mixture-of-Experts (MoE) models (see [1-4] from the reference list below), different token routing strategies can play an important role in the model performance. I am wondering how token routing strategies will affect the proposed method and its analysis. It might be better to include some discussions about it.

**Reference:**
[1] Nguyen, et al. "Statistical Advantages of Perturbing Cosine Router in Sparse Mixture of Experts." arXiv preprint arXiv:2405.14131 (2024).
[2] Liu, et al. "Gating dropout: Communication-efficient regularization for sparsely activated transformers." International Conference on Machine Learning. PMLR, 2022.
[3] Zuo, et al. "Taming sparsely activated transformer with stochastic experts." arXiv preprint arXiv:2110.04260 (2021).
[4] Lewis, et al. "Base layers: Simplifying training of large, sparse models." International Conference on Machine Learning. PMLR, 2021.

**Questions:**

See my comments above.

---

> ### Author Response · Authors · 2025-11-26
>
> Thank you for your thorough reviews and constructive comments. We provide our responses to your comments below and have made major revisions in our revised manuscript. To enhance clarity, we have highlighted the revised text in blue for easy identification.
>
> **A1**: Thank you for the insightful comment. We found that if the model is trained fully end to end from the beginning, then expert specialization does not reliably emerge. The reason is that, in the early iterations, the gradients received by each expert are nearly indistinguishable. Without an initial mechanism to break symmetry, every expert sees almost the same mixed signal and therefore moves in the same direction. As a consequence, the router continues to dispatch inputs in a near random manner, and the system behaves similarly to a single dense transformer. This leads to a convergence rate that matches the multi-head transformer baseline rather than the improved behavior achieved when experts specialize. This observation is consistent with our theoretical characterization of the conflicting gradient regime in Proposition 1.
>
> In contrast, our three stage training view separates the periods during which experts differentiate, attention learns to align with the class specific direction, and the model enters the fast convergence regime. This staged perspective is not intended to prescribe a specific training algorithm. It serves as a conceptual decomposition of the learning dynamics that enables a rigorous theoretical analysis. It explains why specialization arises, when the router stabilizes, and how attention alignment accelerates convergence.
>
> To support this explanation, **we have added a set of experiments (Fig. 5 and Fig. 8)** comparing end to end training with the three stage training dynamics. We also plotted the routing history of end-to-end algorithm (Fig. 6), which shows that the router randomly selects experts, even after convergence. The results confirm our theoretical prediction. End to end training fails to form meaningful expert specialization and converges at a rate similar to the dense transformer. The three stage dynamics successfully separate experts and achieve faster convergence. We have added the above discussion in the revised manuscript.
>
> **A2**: Thank you for the suggestion. We have conducted additional experiments on Tiny ImageNet, which is a significantly larger benchmark than CIFAR level datasets. The corresponding training dynamics are now included in Fig. 9. The specialist formation and attention alignment behaviors match the predictions of our theory and remain consistent with our results on CIFAR, confirming the robustness of the theoretical insights when the data distribution becomes more complex.
>
> Regarding large language benchmarks such as GLUE, MMLU, HellaSwag, and PIQA, we would like to clarify that our theoretical analysis focuses on the training dynamics of Mixture of Transformers, particularly the emergence of expert specialization and attention alignment during training. These phenomena occur in the early and middle stages of optimization. Pretrained large language models, however, are already heavily optimized. They do not exhibit the exploration, specialization formation, or attention alignment processes that our theory characterizes. As a result, evaluating pretrained LLMs does not meaningfully test or validate the theoretical contributions of this work. For this reason, we follow prior theoretical studies on transformer learning dynamics, including Yang et al. 2025 and Huang et al. 2024, and use controlled from scratch training settings where the evolution of specialization, attention alignment, and router stabilization can be observed directly. This choice allows us to faithfully evaluate the predictions of our theory.

---

> ### Author Response · Authors · 2025-11-26
>
> **A3**: Thank you for the thoughtful question regarding routing strategies. [1]-[4] show that routing choices can significantly influence the behavior of sparse MoE systems, and we summarize here how they would possibly interact with the training dynamics analyzed in our work.
>
> - Perturbed cosine routing [1] stabilizes the learning of router parameters by preventing slow convergence of the cosine similarity scores. In our MoT training this type of router would behave similarly to our noisy router early on, but may produce cleaner and more stable partitions once the router moves away from uniformity. This could improve the clarity of specialization and help accelerate the transition from exploration to the alignment stage, although the analysis would need to account for cosine based similarity scores and the perturbation dynamics.
>
> - Gating Dropout [2] introduces controlled randomness into the routing path. In our three-stage training algorithm, it acts as an additional exploration mechanism. Moderate levels of dropout would preserve the conflicting gradient signals that drive specialization, while also reducing early router collapse and improving load balance. High dropout probabilities may slow specialization by making routing too random, but moderate dropout remains compatible with our mechanism. Adjusting the proofs would require modeling this dropout as a modified exploration schedule.
>
> - Stochastic expert routing, such as the approach used in THOR [3], is less compatible with our specialization mechanism. These methods route tokens to experts randomly and often encourage consistency across experts rather than divergence. Such routing suppresses the conflicting gradient effect that is essential in our MoT training, and therefore would prevent or significantly weaken the emergence of specialized experts. As a result, the geometric convergence behavior that we prove after specialization would not hold under stochastic routing of this type.
>
> - Balanced assignment strategies, such as the BASE layer [4], enforce near uniform load balancing across experts before the router stabilizes. In our Stage I setup, this would help maintain stable gradient magnitudes for all experts and may even strengthen the specialization effect by ensuring that each expert consistently receives a meaningful and diverse subset of data. However, this coupling across tokens introduces a global assignment problem that would require a more complex analysis than the simple noisy top one routing we study.
>
> In summary, several routing strategies including balanced assignment, Gating Dropout, and perturbed cosine routing can be integrated into our training framework and may improve load balance or router stability. Fully stochastic routing that enforces expert agreement is in tension with the specialization mechanism that our theory relies on. Our objective in this work is to provide a clear and rigorous first step toward understanding training dynamics of transformer based MoE systems under a general and analytically tractable routing rule. Extending the analysis to more complex routing mechanisms is an interesting direction for future research, and we have included the above discussion in the revised manuscript.
>
> **We hope our responses have addressed your questions. If so, we would be sincerely grateful if you would consider raising your rating to reflect this. Of course, we are happy to answer any further questions you may have. Thank you again for the time and thoughtful attention you have given to our work.**

---

### Official Review · Reviewer_oJET · 2025-10-31

**Soundness:** 2
**Presentation:** 2
**Contribution:** 2
**Rating:** 4
**Confidence:** 3

**Summary:**

This paper proposes MoT, treating an entire Transformer block as a single expert. The primary goal is to establish a theoretical foundation for the MoT architecture.

**Strengths:**

1. To the best of my knowledge, the theoretical foundation of MoT is new.
2. The paper is clear and easy to follow.

**Weaknesses:**

1. While I appreciate the theoretical challenges involved, the proposed model appears somewhat simplified. More importantly, the authors do not sufficiently address the gap between theory and practice. For instance, do the optimal choices of \( T_1 \) and \( T_2 \) suggested by Propositions 1 and 2 actually translate into practical improvements? Furthermore, Figure 2 does not demonstrate the expected linear convergence rate—performance remains comparable to baselines that only enjoy sublinear theoretical convergence. The paper would benefit from a deeper discussion of why this discrepancy occurs and what strategies could help bridge the theory–practice gap.

2. The experimental evaluation is relatively limited. At a minimum, the authors should include comparisons against standard MoE architectures. Additionally, to strengthen the empirical validation and better support the theoretical claims, experiments with larger-scale models—such as LLMs or MLLMs—and a broader range of tasks (e.g., vision, language, and multimodal benchmarks) would significantly enhance the paper’s impact and credibility.

3. It appears that the MoT architecture currently supports only sequence-level routing. This design choice may lead to suboptimal load balancing across experts and could limit computational efficiency, especially in settings where token-level sparsity is beneficial. The authors should discuss the implications of this limitation and consider whether finer-grained routing mechanisms could be incorporated in future work.

**Questions:**

See above

---

> ### Author Response · Authors · 2025-11-26
>
> Thank you for your thorough reviews and constructive comments. We provide our responses to your comments below and have made major revisions in our revised manuscript.
>
> **A1**: Thank you for highlighting these important points. We address such aspect below:
>
> (1) **On the MoT model**: Our considered one-layer transformer model allows for a tractable yet insightful understanding of the core learning dynamics in Mixture-of-Transformers (MoT) architectures. This abstraction is consistent with prior theoretical work on transformers (e.g., Yang et al. (2025), Huang et al. (2024), Chen et al. (2022)) and serves as a foundational step toward understanding deeper and more complex models.
>
> Even in this single-layer setting, the interaction between attention layers, FFNs, gating, and routing **presents significant technical challenges**. By isolating these components, we are able to prove specialization, attention alignment, and fast convergence, which guarantees that remain out of reach for full multi-layer transformer stacks under MoT routing. Importantly, the derived results also provide fundamental insights on practical multi-layer transformer systems. Importantly, our empirical results (Figure 2) demonstrate that the core insights from this analysis (expert specialization, noise suppression through attention, and improved convergence) continue to hold in practice when applied to multi-layer ViTs, indicating that the simplified model captures key mechanisms that transfer to realistic architectures.
>
> (2) **On the relation between $T_1,T_2$ and practice**: The theoretical expressions $T_1=O(\eta^{-1}\sigma_0^{-0.5}M)$ and $T_2=T_1+O(\eta_a^{-1}\sigma_0^{-0.5}N^{-1}M)$ are upper bounds ensuring specialization (Prop. 1) and attention alignment (Prop. 2). They are not intended as optimal choices in practice. In experiments, we used these expressions as guidance for initial values and refined them based on the training-loss dynamics. The empirical sensitivity study included in Appendix (Figure 8) further shows that the model is robust to a broad range of choices of $T_1$ and $T_2$, as long as they exceed the minimum threshold needed for specialization and attention learning. However, choosing values that are too small can hinder convergence, especially in more complex datasets. We have integrated these insights more clearly in our revised manuscript to provide practical guidelines for practitioners.
>
> (3) **On Figure 2**: We clarify that the linear convergence rate $O(log(\epsilon^{-1}))$ in our theory refers to geometric decay of the error once specialization and attention alignment have been achieved after $T_2$. Before this point, the loss decreases slowly due to the exploration and router stabilization predicted by Proposition 1. Figure 2 shows the entire training trajectory, so the early slow phase visually dominates. When focusing on the post specialization window, the MoT model exhibits a sharper error drop than the multi head transformer.
>
> In summary, although the theoretical model is simplified, the captured core mechanisms, including expert specialization, attention-driven noise filtering, and post-specialization acceleration, manifest robustly in practice. Meanwhile, the theory provides actionable guidance: (1) ensure sufficient early-stage exploration via $T_1$, (2) allow attention refinement before fine-tuning via $T_2$, and (3) expect the fastest convergence after specialization stabilizes. We have made these practical insights and the relationship between theory and empirical behavior more explicit in the revised manuscript.
>
> **A2**: Thank you for the suggestion. We have added comparisons against standard MoE architectures to the Appendix (Fig. 4), where we compare MoT with both FFN-level MoEs and multi-head transformers on CIFAR-100. The results consistently show that MoT outperforms standard MoE baselines, further supporting the benefit of attention alignment to the expert specialization. We also clarify why our main text highlights the comparison with attention-absent MoE. This baseline is essential for isolating the contribution of the attention mechanism itself, which is a central component of our theoretical analysis. The stark performance gap between attention-absent MoE and MoT directly reflects Lemma 1, and thus provides the clearest empirical validation of the theoretical claims.
>
> Regardin large-scale LLMs: we focus on the training dynamics of MoT, especially the specialization and attention alignment emerge during training. Pretrained LLMs are already deeply optimized and do not expose the early-stage behaviors (exploration, specialization, attention alignment) that our theory predicts. As a result, using pretrained LLMs does not meaningfully validate the theoretical contributions of this work. We therefore follow prior theoretical studies (e.g., Yang et al., 2025; Huang et al., 2024) and evaluate on controlled, from-scratch training settings where the dynamics can be faithfully observed.

---

> ### Author Response · Authors · 2025-11-26
>
> **A3**: Thank you for raising this important point. The sequence-level routing used in our paper follows the theoretical MoE literature (e.g., Chen et al., 2022, Li et al. 2025) and allows us to obtain clean analytical guarantees on specialization, attention alignment, and convergence. Finer-grained (e.g., token-level) routing introduces substantially more stochasticity and coupling between attention patterns and the router, making the training dynamics much more difficult to characterize theoretically.
>
> Regarding load balancing, we note that our use of normalized gradient descent in Stage I plays a crucial role in promoting expert exploration and mitigating routing imbalance, which is similar to the mechanism used in Chen et al. (2022). This normalization ensures that all experts receive meaningful gradient signals during early training, preventing collapse to a small subset of experts even under sequence-level routing. Empirically, Figure 3 demonstrates that the router achieves effective specialization without severe imbalance on CIFAR-10/100.
>
> We also emphasize that our NLP experiments in the Appendix (e.g., Figs. 10-12) show that the insights derived from our analysis (e.g., expert specialization, attention alignment, and accelerated convergence) extend beyond vision data to more general sequence tasks, further supporting the practicality of sequence-level routing in a variety of domains.
>
> We agree that extending MoT to finer-grained routing (e.g., token-level or patch-level routing) is an exciting direction for future work. Such mechanisms could potentially improve computational efficiency by enabling token-wise sparsity, but they introduce significantly more complex interactions between routing, attention, and expert specialization. **Our current work provides the first step toward understanding the training dynamics of Mixture-of-Transformers**, offering provable characterization of specialization and convergence under a tractable sequence-level routing scheme. We view this as an essential theoretical foundation upon which future analyses of more expressive routing mechanisms can be built. We will add this discussion to the revised manuscript.
>
> **We hope our responses have addressed your questions. If so, we would be sincerely grateful if you would consider raising your rating to reflect this. Of course, we are happy to answer any further questions you may have. Thank you again for the time and thoughtful attention you have given to our work.**

---

### Meta-Review · Area_Chair_o6sJ · 2026-01-04

**Summary:**

This paper studies a Mixture-of-Transformer model where each transformer block acts as a single expert. The reviewers had the following concerns:
1. Lack of empirical studies on large models and on NLP tasks with standard benchmarks; The reviewers think these more realistic experiments would strengthen the theoretical claims and potentially show the practical utility of the model and the theory.
2. The proposed model is trained with a three-stage algorithm (train some component and freeze others in each stage); The reviewers ask additional comparison against models trained with standard end-to-end procedure.
3. Overly simplified model (single-layer transformer), and strong data assumption (data orthogonality); The reviewers are unclear whether the theoretical insights (multi-phase training dynamics, efficiency improvement) are still valid for more realistic models.
4. Some reviewers concern about the lack of novelty when compared with Chen et al. 2022. They also concern about potential unfair comparison in the experiments.

**Reviewer Concerns:**

1. For the first concern, the authors clarified that this paper is more theory-focused thus the experiments are for validating the theoretical claims. I think the point the author made in the clarification is reasonable and the added experiments further validate the theoretical claims. However, there is still no experiments with larger models and more realistic tasks in other domains. With three reviewers demanding those empirical validations, I view this concern still outstanding. In my own view, since this paper proposed a relatively new mixture of expert design, it is also necessary to at least confirm some of its practical value (in realistic settings) before one can fully appreciate the theoretical results on these model. Therefore, lack of empirical validation remains to be a major weakness of this paper.

2. For the second concern, the authors provided additional experiments with MoTs trained by standard procedures, showing a performance drop compared to those with three-stage training. I think this concern is mostly addressed. However, I have checked those experiments and notice that all the multi-head transformer and MoE baselines are trained by the proposed three-stage algorithm, which is unjustified. A fair comparison should be with those models trained with their respective standard training procedures, as there is no evidence that the multi-stage algorithm would benefit those models.

3. For the third concern, as the the criticism in my view is not the simplicity of the model per se, but rather the limited practical implications of the theoretical results, which should have been addressed by empirical experiments on realistic settings. Since the empirical validations remain insufficient after rebuttal, I view this point still outstanding.

4. The forth concern is addressed by the rebuttal.

**Reviewer Scores:**

I think the reviewers oJET (4), dwgC (6) would keep their score. It is possible for reviewers Ru49 to increase their score (4 to 6), depending on how they think of the additional experiments. Reviewer dCo3 would have changed their score (2 to 4) since their primarily concern about comparison to prior works and inappropriate baselines is addressed.

Given the borderline scores and limited empirical results in this paper, I have to recommend rejection.

---

### Decision · Program_Chairs · 2026-01-26

Reject